# SYNAPSE: SImulation Benchmark of Neuro-Adaptive Patient-Specific Evaluation for Episodic Decision-Making

## Abstract

Recent advances in time-series analysis, treatment outcome prediction, and reinforcement learning (RL) have demonstrated great potential to automate decision-making in healthcare. However, the high stakes nature complicates the deployment of such frameworks in practice, clinically, or in the long term. A major challenge is the absence of realistic benchmark environments that capture the sequential, patient-specific nature of various therapies, which could enable extensive offline testing, evaluation, and model selection prior to clinical adoption. To address this, we introduce the SImulation Benchmark of Neuro-Adaptive Patient-Specific Evaluation (SYNAPSE), in the context of adaptive deep brain stimulation (DBS), a treatment for managing the motor symptoms of Parkinson's disease (PD). Specifically, SYNAPSE is constructed using real-world data collected from both clinical and at-home studies involving participants undergoing DBS therapy. It enables **offline training and evaluation** of different treatment strategies, reflecting both short- and long-term effects, as well as **treatment outcome prediction** capturing participants' responses to a range of temporal dynamics. Additionally, it allows for the assessment of safety-critical constraints inherent to neurostimulation decision-making. By rigorously validating its realism against clinical data and supporting both short- and long-term decision-making, SYNAPSE offers clear guidance for future DBS policy development, as well as helps identify and address key challenges in advancing truly personalized neurostimulation therapies.

## 1 Introduction

Healthcare often involves making a series of personalized treatment decisions and refinements over time, *i.e. episodic* decision-making, which can be roughly categorized into two different types. **Decision Type *(I)* – *high-level episodic decision-making at a slower pace* – typically occurs during clinical visits, where clinicians continually re-evaluate the interventions and dosages to prescribe at each stage, taking into account the patient's evolving condition and historical responses (Denton, 2018). These decisions provide overall guidance for ongoing treatment. **Decision Type *(II)* – *low-level episodic decision-making at a faster pace* – for diseases where implantable devices are pivotal, *e.g.*, deep brain stimulation (DBS) for Parkinson's disease (PD), epilepsy and autism, or implantable cardioverter-defibrillators for ventricular tachycardia, decisions must be made every few seconds or less and with greater granularity. Data-driven methods are commonly employed to optimize treatment for both types. For decision type *(I)*, counterfactual prediction and causal models are commonly used to inform clinicians about likely outcomes under different interventions in the longer term (Zammel et al., 2024; Andersen et al., 2023; Ho et al., 2008), *e.g.*, for forecasting patient readmission risk or disease progression. For decision type *(II)*, frameworks like Markov decision processes (MDPs) and stochastic modeling are used to formalize the rapid, sequential decision problem under uncertainty, *e.g.*, for designing controllers of implantable medical devices (Denton, 2018; Prasad et al., 2017; Raghu et al., 2017). In combination, these approaches leverage routinely collected clinical data to forecast patient trajectories and inform personalized therapies, ultimately aiming to improve outcomes and safety in long-term.

Despite this promise, healthcare currently lacks realistic, data-driven environments for safely training and validating algorithms before deployment for clinical and daily at-home use. Safety, privacy,

ethical, and practical constraints make it impossible to train *online* policies directly with patients, narrowing the options to relying on counterfactual data or simplistic models (Thomas & Brunskill, 2016; Gao et al., 2023b). For example, patients affected by PD, the 2nd most common neurode-generative movement disorder (Ben-Shlomo et al., 2024), may no longer respond well to drugs and thus need DBS as a supplementary therapy (Armstrong & Okun, 2020). Adaptive DBS (aDBS) systems dynamically adjust stimulation intensity based on signals capturing neuronal feedback such as local field potentials (LFPs) to optimize symptom control, but the invasive nature of aDBS makes it impractical to collect large-scale patient data or run cohort-based randomized experiment with new control strategies *in vivo* (Gao et al., 2020). Existing computational models of DBS are mostly indicative and remain immature in their ability to simulate feedback from real patients, in the sense that they require the knowledge to the level of neuronal firing patterns for every single neuron in the affected area of the brain (Cho et al., 2024; Kuzmina et al., 2025; Pan et al., 2024), making it intractable to be personalized according to disease's and individual's characteristics. In contrast, clinical and at-home experimental data are usually considered proprietary due to privacy agreements and institutional review board (IRB) limitations. Such gaps highlighted the needs for realistic healthcare simulators that can serve as virtual patients to design, evaluate and constantly refine the treatment strategies as needed.

In this work, we introduce the SImulation benchmark of Neuro-Adaptive Patient-Specific Evaluation (SYNAPSE), an open-source simulator built on extensive real-world DBS data collected from PD participants undergoing a variety of treatment strategies. It can facilitate use cases under both decision types *(I)* and *(II)* introduced above. Specifically, SYNAPSE leverages longitudinal recordings from participants with PD where each received 4 different types of DBS therapy, under both clinical and at-home setups. Moreover, each participant displayed unique disease characteristics (*e.g.*, among tremor, dyskinesia, bradykinesia, gait imparment), ensuring the heterogeneity within the dataset. By training a generative model, via variational auto-encoding (Gao et al., 2023b), the distributions of state transitions and rewards/human feedback are captured, with the participants' characteristics encoded into the latent prior from which the simulated trajectories are generated upon. As a result, SYNAPSE expresses various disease and participant-specific characteristics and effectively functioning as virtual patients *in silico*, *i.e.*, clinicians or algorithms can input a proposed DBS policy and instantly observe the simulated interactions and outcome over time, facilitating the training, evaluation, refinement as well as predicting the outcome of the personalized treatment policies under both decision types *(I)* and *(II)*.

Our goal is to bring policy learning and evaluation closer to real-world clinical practice by providing the community with access to a clinic-like environment. This enables the systematic development and assessment of effective, patient-specific episodic decision-making, and advancing the person-alization of treatments without being constrained by clinical deployment bandwidth or compromis-ing data privacy. Specifically, the contributions of SYNAPSE are summarized as (*i*) introducing a robust simulation framework in the context of train, evaluate, and refine/finetune aDBS control policies (*i.e.*, decision type *(II)*), as well as long-term outcome/patient feedback predictions (*i.e.*, de-cision type *(I)*), (*ii*) rigorously validating the simulator's ability to model state transitions and issue rewards/human feedback using policies trained on clinically collected data (from real participants), with its effectiveness benchmarked against extensive clinical and at-home testing results, and (*iii*) benchmarking leading reinforcement learning (RL) algorithms (both online and) in a unified, clini-cally relevant setting, providing clear guidance for future DBS policy development and deployment. Moreover, as the aDBS control problem still remains as an open question, we hope the release of SYNAPSE could help uncover key challenges in a more realistic setup, as well as and drive the development of both training and evaluation algorithms toward delivering truly patient-specific ex-periences.

## 2 RELATED WORKS

### 2.1 EXISTING HEALTHCARE SIMULATIONS FOR EPISODIC DECISION-MAKING

Healthcare simulators have in general played a pivotal role in domains other than DBS (*e.g.*, (Jiang et al., 2025; Fan et al., 2025; Schmidgall et al., 2024; Liventsev et al., 2021; Theodorou et al., 2025)). Specifically, MedAgentBench (Jiang et al., 2025) created a virtual electronic health record (EHR) environment with 100 patient profiles for testing large language model (LLM) agents on a

variety of clinical tasks (*e.g.*, records retrieval and aggregation, as well as test and referral ordering) through conversations with the agent. MediSim provided a generative framework that simulates realistic patient trajectories by filling in missing visits and data modalities in EHRs (Theodorou et al., 2025). Such simulators mostly aim to facilitate the decision type *(I)* (see Section 1), providing analytical insights and potential treatment options for clinicians as references. Similarly, several other healthcare simulators have been developed for prompting large language models (LLMs) in doctor-patient interactions and question answering tasks (Pal et al., 2022; Singhal et al., 2025).

There also exist simulators that focus on extrapolating average treatment effects across cohorts using counterfactual causal modeling with historical data, *i.e.*, not specific to an individual or a sub-group of patients, for sepsis (Oberst & Sontag, 2019). On the other hand, empirical data have also been used (indirectly) to facilitate expressive mathematical modeling of organ/disease dynamics, which lead to simulations *in silico*, *e.g.*, for diabetes (Man et al., 2014) and PD (Kuzmina et al., 2025). Similarly, there also exist simulators developed as digital twins (Geddes et al., 2025), which captures the hemodynamic changes in the pulmonary arteries where Markov Chain Monte Carlo can be used to identify predictive markers for heart disease progression and outcomes forecasting. Although these types of simulators could provide more granular pathological insights, *i.e.*, closer to facilitate decsision type *(II)*, they generally lack the ability to capture and model disease characteristics specific to individual patients.

In contrast, SYNAPSE is designed to serve as the testbed for both decision types *(I)* and *(II)*, while reflecting the unique characteristics from individuals across the cohort.

### 2.2 BRAIN NEURONAL ACTIVITIES MODELING AND aDBS CONTROL

Existing commercial DBS devices only provides open-loop continuous DBS (cDBS) stimulation with fixed parameters (usually set by clinicians from trial-and-error) (Pineau et al., 2009). In contrast, adaptive DBS (aDBS) for Parkinson's disease is an emerging closed-loop therapy where stimulation parameters are adjusted on the fly based on patient state (e.g. neural signals or symptoms), aiming to control PD symptoms while conserving the battery life of the implantable pulse generator (IPG) (Hoang et al., 2017). Given the high-stake nature and cost for recruiting participants willing to be implanted with custom aDBS devices (mostly approved by exceptional and experimental use only), most existing aDBS algorithms are tested with numerical experiments leveraging biophysical brain models derived from animals (So et al., 2012; Jovanov et al., 2018; Kumaravelu et al., 2016; Pan et al., 2024). Specifically, Gao et al. (2020) introduced an RL framework using beta-band power spectral density of local field potentials (LFPs) from the basal ganglia (BG) as the environmental states, and the stimulus temporal patterns as the action space. The algorithm was tested on a hardware implementation of the BG model captured as ordinary differential equations introduced by Jovanov et al. (2018). The results showed that the RL framework was able to use 70% less battery consumption to control PD symptoms comparable to cDBS. Other examples of RL and other aDBS innovations tested on similar testbeds include Cho et al. (2024); Mnih et al. (2015); Gu et al. (2017); Beudel & Brown (2016); Arlotti et al. (2016b;a); Little et al. (2016b); Swann et al. (2016); Opri et al. (2020); Pan et al. (2024). For example, Coprocessor Actor Critic work presents a model-based RL approach for adaptive brain stimulation via an in silico model of brain injury (Pan et al., 2024). However, translating these findings to real-world aDBS is challenging, as such testbeds approximate DBS responses based on animal data and cannot be directly extended to real patients or capture the diverse characteristics and symptoms of PD across individuals.

There also exist aDBS controllers validated through clinical testing. For example, Schmidt et al. (2024) developed a proportional-integral (PI) controller with its effectiveness verified through recurring clinical sessions. Gao et al. (2023c) tested RL-based stimulation amplitude adapters in the clinic and found human feedback received are comparable to that of cDBS. Gilron et al. (2021) introduced an aDBS controller that switches between two modes by detecting if the patient is asleep or awake. However, the data collected from such studies are typically not openly accessible, making it difficult to reuse them for training new controllers or conducting counterfactual analyses and evaluations. Also, prior work (Gao et al., 2023c) lack investigations over cohort-generative modeling with long-term effects such as human feedback. Consequently, there is a clear gap between the needs of machine learning (ML), RL, and control practitioners for an environment that enables extensive evaluation and improvement of algorithms, ultimately providing greater confidence in advancing innovations that enhance human outcomes.

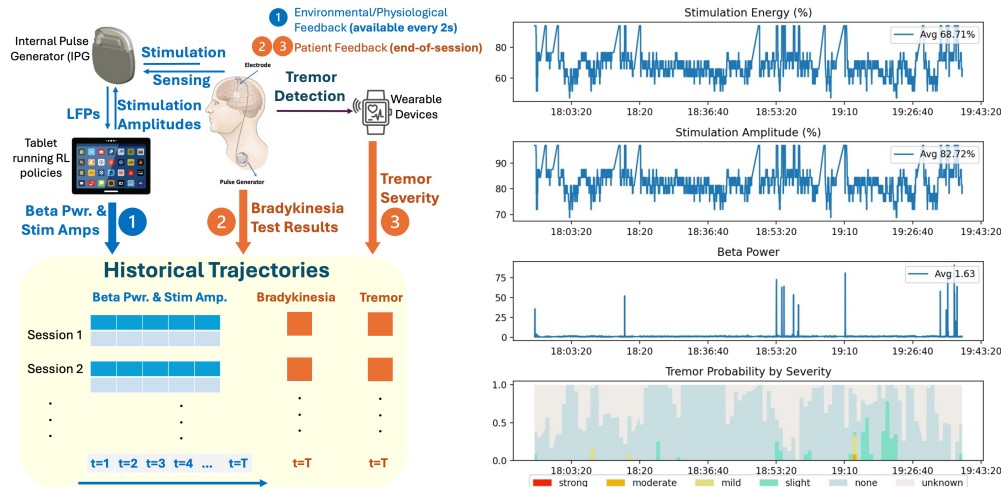

Figure 1: (**Left**) Setup of the neurostimulation experiments and the formulation of collected trajectories. Environmental rewards and human returns are captured in streams 1 and 2-3 respectively. (**Right**) An example of a recorded session of a patient, including the stimulation energy and amplitude, beta power, and the tremor probability by severity.

## 3 THE DBS DATASET USED TO DEVELOP SYNAPSE

As illustrated in Figure 1, the dataset used to develop SYNAPSE is collected from real-world clinical and at-home testing of aDBS therapy over 5 participants with PD; testing period is from April 2022 to May 2025 and detailed stats can be found in Table 2[1]. The implantable pulse generator (IPG) was implanted to the participants, which can both stimulate and sense local field potentials (LFPs) from the basal ganglia (BG). The IPG streams LFP data (sampled at 500 Hz) to a tablet, which computes beta-band power (by applying FFT to calculate the power spectral density from 13-35 Hz) as the neural biomarker. The aDBS controller determines the stimulation amplitude (0-100%) every 2 seconds based on the latest beta power. Both sensing and stimulation are facilitated by the 4-contact electrodes implanted to the subthalamic nucleus (STN)/globus pallidus, delivering monopolar stimulation from a single contact while sensing from adjacent once, *i.e.*, the *sandwich sensing*.

**Participant Characteristics.** Participants' characteristics varied widely in terms of PD symptoms (*i.e.*, tremor, bradykinesia, gait impairment, etc.) as well as the Levodopa equivalent daily dosage (LEDD), which can be found in Table 1. For example, participant #1 only experiences episodes of tremor, and has Bradykinesia in the left hand which correlates with the beta power computed from LFPs collected from right STN. Participant #3 has very large amplitude tremor returns within

Table 1: Characteristic of each participant, including the level of tremor when receive insufficient stimulation, if affected by bradykinesia as a symptom of PD, as well as the PD medication dosage quantified as Levodopa equivalent daily dosage (LEDD) (Julien et al., 2021).

|  | Tremor Level | Bradykinesia | Gait | LEDD (mg) |
|---|---|---|---|---|
| Participant #1 | Sporadic | Yes (Left Hand) | No | 57 |
| Participant #2 | Constant | Yes (Right Hand) | No | 113 |
| Participant #3 | Constant | Yes (Right Hand) | No | 200 |
| Participant #4 | Constant | Yes (Right Hand) | No | 100 |
| Participant #5 | None | Yes (Right Hand) | Yes | 713 |

seconds after being treated with stimulus with insufficiently low amplitudes, and bradykinesia in right hand highly correlated with left STN beta power. Accordingly, SYNAPSE models each patient as an individual environment, reflecting the *virtual* cohort of participants displaying similar characteristics, rather than pooling all data. This preserves personalized dynamics and enables testing how well a policy trained on one patient generalizes to another.

**Clinical Sessions.** Each participant visits the clinic once every 1-2 months. Four types of aDBS policies were deployed to each participant in the clinic. Specifically, the policies include continuous DBS (cDBS), turning off DBS (DBS-OFF), proportional-integral (PI) (Wang et al., 2016), and RL. cDBS always stimulates at the highest allowed amplitude (determined by clinicians) at all time,

---

[1]Written consents obtained from all participants. IRB approved from the university and health system where experiments were conducted.

and DBS-OFF completely turns off DBS within a short window (the IPG output is set to zero for that period); both are considered open-loop control. The PI and RL controllers perform closed-loop aDBS by adapting the stimulation amplitude according to the latest beta power readings, at every 2 seconds. The PI controller's parameters are tuned through trial-and-error jointly by control engineers and clinicians. The RL controllers are trained using deep deterministic policy gradient (DDPG) (Lillicrap et al., 2016) with data collected from three other types of controllers, followed by finetuning with latest data. Each clinical session usually lasts 5-20 minutes, and human feedback were collected at the end of each session quantifying the level of bradykinesia (*i.e.*, hand-grasp speed by recording the number of making-fist-and-release maneuver the participant can perform within 10 seconds) and tremor (fraction of session with detected tremor via wrist accelerometer) during the session. The top half of Table 2 documents the total number of sessions/episodes and state transitions provided by all sessions, total number of policies deployed clinically, the range, mean and std of session lengths, as well as the total time spent testing all policies in the clinic.

**At-Home Sessions.** Two participants opted in for testing aDBS at home, where they can choose when and where to start and stop the sessions. Only 2 types of policies were deployed (*i.e.*, one RL and cDBS) in this setup – when the participant chose to start a session, one of the three controllers was uniformly randomly chosen to start until the participant chose to end the session. The tremor feedback is available at the end of each session, while the bradykinesia evaluation is not. The bottom part of Table 2 shows the overall volume of the at-home dataset.

Table 2: Statistics of Collected Data from Each Cohort. The dimension of observations at each step is 10, action space is 1 with a continuous value, respectively.

| Participant # | 1 | 2 | 3 | 4 | 5 |
|---|---|---|---|---|---|
| Clinical DBS Dataset | | | | | |
| Total Sessions | 122 | 156 | 208 | 192 | 165 |
| Total State Transitions | 32K | 40K | 47K | 49K | 40K |
| Total # of Policies | 6 | 8 | 13 | 6 | 6 |
| Session Steps Range | 21-437 | 14-587 | 19-437 | 47-585 | 19-582 |
| Session Steps Mean (Std) | 261 (153) | 259 (169) | 226 (128) | 254 (152) | 237 (160) |
| Total Clinical Time (min) | 1714 | 1757 | 1138 | 1515 | 1245 |
| Observed Human Rewards Mean (Std) | 215 (34) | 143 (39) | 139 (34) | 175 (51) | 232 (36) |
| At-home DBS Dataset | | | | | |
| Total Sessions | - | 59 | - | 192 | - |
| Total State Transitions | - | 36K | - | 476K | - |
| Total # of Policies | - | 2 | - | 2 | - |
| Session Steps Range | - | 4-1798 | - | 5-22741 | - |
| Session Steps Mean (Std) | - | 605 (511) | - | 2481 (4307) | - |
| Total At-Home Time (min) | - | 1189 | - | 15878 | - |

**Challenges for ML/RL in DBS.** As noted from prior aDBS works (Gao et al., 2020), the invasive nature of aDBS, as well as the need for custom DBS devices (Schmidt et al., 2024; Gilron et al., 2021; Gao et al., 2023c), makes it challenging and costly to recruit participants, making randomized experiments and large-scale data collection intractable. As a result, although RL and other aDBS approaches have been explored, most of the studies remain with numerical verifications (Cho et al., 2024; Mnih et al., 2015; Gu et al., 2017; Gao et al., 2022; Beudel & Brown, 2016) in environments that are arguably oversimplified and lack the patient-specific nuances present in *in vivo* experiments, making it difficult to translate theoretical and algorithmic advances into clinical practice. Moreover, even when *in vivo* DBS experiments are available at specific institutions, privacy agreements and IRB regulations often hinder data sharing with the public or the ability to conduct extensive experiments for external requests. As a result, most ML and RL researchers have very limited opportunities to advance new algorithmic research in aDBS.

## 4 THE SYNAPSE

SYNAPSE addresses the challenges above by capturing and learning patient-specific dynamics in responses to various aDBS treatments from the clinical and at-home data introduced above. It enables safe and efficient training and evaluation of policies with the potential to improve treatment decision-making in future patient trials.

**Environmental Setup** We formulate the patient-specific DBS environment as a human-involved MDP (HMDP), which is a tuple $\mathcal{M} = (\mathcal{S}, \mathcal{A}, \mathcal{P}, R, R^{\mathcal{H}}, s_0, \gamma)$, where $\mathcal{S}$ is the set of states, $\mathcal{A}$ the set of actions, $\mathcal{P} : \mathcal{S} \times \mathcal{A} \to \mathcal{S}$ is the transition distribution usually captured by probabilities $p(s_t|s_{t-1}, a_{t-1})$, $R : \mathcal{S} \times \mathcal{A} \to \mathbb{R}$ is the *environmental* reward function, $R^{\mathcal{H}}(r^{\mathcal{H}}|s, a)$ is the *human* reward distribution from which the immediate human reward (IHR) $r_t^{\mathcal{H}} \sim R^{\mathcal{H}}(\cdot|s_t, a_t)$ are sampled, $s_0$ is the initial state sampled from the initial state distribution $p(s_0)$, and $\gamma \in [0, 1)$ is the discounting factor. Specifically, the immediate environment reward $r_t = R(s_t, a_t)$ is a deterministic function combining the change in beta-band power and stimulation amplitude, while the immediate human reward (IHR) is a stochastic feedback signal drawn from a learned distribution $R^{\mathcal{H}}$ at each time step. Thus IHR captures short-term patient responses (e.g. tremor/bradykinesia outcomes) that are not captured by beta power alone. Moreover, the IHRs are usually considered partially observable,

Figure 3: Schematic of SYNAPSE and Pipeline for Supported Tasks.

or unobservable whereas the end-of-episode human return, $\sum_t \gamma^t r_t^{\mathcal{H}}$, is fully observable. The aDBS controller interacts with the HMDP following some policy $\pi(a|s)$ that defines the probabilities of taking action $a$ at state $s$. In typical aDBS setups (Gao et al., 2020), the immediate rewards $R$ is a combination of beta power change (lower the better) and stimulation amplitude usage (lower the better; to conserve the battery life of the IPG); the aDBS policy is then trained to maximize the mixture of immediate rewards and IHRs.

Following from Section 3, the states are set to be the historical beta power values within the past 20-second window. The action is the percentage of the maximum amplitude allowed per participant (which is determined by clinicians). Each state-action transition corresponds to a 2-second control cycle where the controller updates the stimulation amplitude once within. The immediate rewards are defined based on the trend of beta power changes, while human feedback is available at the end of each episode, *i.e.*, a combination of bradykinesia and tremor outcomes for clinical sessions, and tremor outcomes only for at-home sessions. More details can be found in Appendix D.4.

**DBS Environment Modeling** As introduced in Section 3, each participant exhibited distinct characteristics and responses to interventions. To account for this heterogeneity, we model each participant respectively as an independent environment, rather than having a single environment representing all participant. Each environment could serve as a virtual cohort, representing patients with similar symptoms and characteristics, thereby preserving patient-specific dynamics and avoiding a one-size-fits-all approach that might overlook important individual differences. This approach also prevents the introduction of irrelevant information that could arise from pooling all patient data together. To this end, SYNAPSE allows users

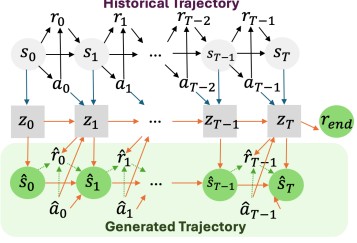

Figure 2: Architecture of the Variational Latent Model with Human Returns per Virtual Cohort.

to examine how well a policy learned on one patient generalizes to others, emphasizing the importance of adaptable strategies across diverse patient profiles. We specifically leveraged a variational auto-encoding technique adapted to capture state-action temporal transitions as well as rewards, IHRs and end-of-episode human returns (illustrated in Figure 2); details can be found in Appendix E.

## 4.1 Task Design with SYNAPSE

Figure 3 illustrates schematic of the aDBS simulation and pipeline for supported tasks with SYNAPSE. It supports diverse learning paradigms, facilitating the robust development and evaluation of patient-specific neurostimulation strategies. It enables training personalized stimulation policies $(i)$ by directly interacting with the environments/virtual cohorts through an *online* manner, or $(ii)$ over collecting a set of *offline* data using pre-determined behavioral policies through an offline manner. Candidate policies can be then evaluated by $(i)$ Markove Chain Monte Carlo (MCMC) as *online* inference, representing the clinical testing sessions, or $(ii)$ through *off-policy* evaluation applied to the hold-out part of offline trajectories, simulating the scenario of offline testing and model selection before clinical sessions. Supporting these various training and evaluation scenarios helps bridge the gap to real-world validation, encompassing both decision types $(I)$ and $(II)$; see Section 1. Throughout the pipeline, human-in-the-loop feedback and long-term patient outcomes can be incorporated to guide policy refinement and ensure clinical relevance. Moreover, the inclusion of diverse cohorts enables zero- or few-shot transfer learning, promoting the development of more robust and efficient algorithms for unseen patients.

### 4.1.1 OFFLINE TASKS

**Offline Treatment Policy Learning.** A fundamental use of SYNAPSE is to train adaptive DBS policies entirely offline, mitigating the risks and meeting constraints of direct patient experimentation Gao et al. (2020; 2023b). In this case, an RL agent can interact with a high-fidelity virtual patient to learn optimal aDBS control strategies. At each step, the RL agent observes a state that encodes recent neural signals (*i.e.*, a history of beta power spectral densities extracted from LFP readings) and relevant symptom indicators, and then selects an action in the form of a percentage stimulation amplitude. Then, SYNAPSE returns the next state and an immediate reward, where the reward function balances symptom suppression and energy efficiency—penalizing excessive amplitude use and failure to reduce pathological beta oscillations. This setup mirrors the closed-loop aDBS setting (state: neural biomarkers like beta power; action: neurostimulator amplitude; reward: reduced beta power with minimal energy usage). Standard deep RL algorithms can be deployed in this offline simulator to optimize policies, iteratively improving control performance without any *in vivo* trials. Moreover, SYNAPSE is also able to simulate the end-of-episode human return (*i.e.*, the discounted sum of unobserved IHRs) for the RL agent to capture and optimize over. The patient-specific nature of the five distinct DBS virtual cohorts also supports transfer learning and zero-shot learning approaches, enabling robust generalization to new patients.

**Off-Policy Evaluation/Selection.** Beyond training, the simulator enables rigorous off-policy evaluation (OPE) of candidate DBS controllers using only historical data. This is a critical step before clinical testing in practice, where each new policy must be vetted for safety and efficacy without trial-and-error on participants. SYNAPSE supports OPE by replaying candidate policies on representative patient scenarios *in silico*. By off-policy evaluation, we intend the scenario of evaluating a policy using only historical/simulated data, as supported by our simulator. Specifically, given a policy, whether a deep RL controller or a clinician-designed heuristic, the simulator can generate rollouts or utilize its latent dynamics model to estimate the policy's performance on previously unseen state-action sequences. This capability is essential for benchmarking OPE algorithms and assessing their ability to handle distribution shifts between offline data and online interactions. In what follows, SYNAPSE yields both standard RL metrics (e.g., cumulative reward) and estimates of end-of-episode *human return* by modeling long-term patient outcomes. For instance, the environment can project tremor severity and forecast patient-reported satisfaction using its integrated human feedback model. This allows for policy selection based on both biomarker-driven efficacy and human-centric outcomes, filtering out suboptimal or unsafe strategies before clinical deployment.

### 4.1.2 ONLINE TASKS

**Policy Testing.** SYNAPSE supports online evaluation of DBS controllers by simulating real-time interactions between the aDBS policy and the virtual cohorts. Specifically, a learned agent is deployed to control the virtual patient in a live feedback loop, mimicking an actual clinical session. SYNAPSE yields an observation every step ( *i.e.*, 2 seconds) from which the policy selects a stimulation amplitude for the next step. SYNAPSE then updates the virtual cohort's internal state and issues rewards reflecting symptom change and stimulation efficiency. This setup enables *in silico* stress-testing and verification of the behavior of a policy over time, for example, to detect failure modes such as high-variance amplitude oscillations. Performance metrics such as cumulative reward, tremor suppression, and energy consumption are recorded to evaluate policy robustness. This online interaction complements offline metrics by revealing emergent behaviors and facilitating safe debugging of DBS strategies prior to real-world deployment.

**Human Feedback Alignment.** A key challenge in RL-driven aDBS is ensuring that the agent's behavior aligns with patient preferences and subjective well-being, not merely neural biomarkers. Our simulator facilitates *RL to maximize human feedback* by incorporating both objective environmental rewards and subjective human returns (*i.e.,* the discounted sum of unobserved IHRs). The simulator includes probabilistic human feedback mechanisms that generate immediate and episodic human returns based on real-world clinical proxies (*i.e.*, hand grasp speed and tremor duration). These feedback channels enable training of agents using composite rewards that combine neural and behavioral indicators. RL algorithms can be trained to optimize these human-aligned returns,

or more specialized methods such as inverse RL (or reward modeling) and policy optimization from human preferences could be developed.

### 4.1.3 HYBRID TASKS

**Long-Term Outcome Prediction.**  In aDBS, the effectiveness of a policy extends beyond short-term symptom relief to long-term clinical outcomes. SYNAPSE supports this by enabling long-horizon rollouts, allowing for *long-term outcome prediction* of candidate policies. Leveraging generative models, SYNAPSE can produce extended sequences of beta power trajectories, stimulation amplitudes, and patient returns. Outcomes such as total energy consumed, tremor time ratio, bradykinesia test results, and final human return can be aggregated to reflect long-term impact. This approach combines offline modeling from logged data with prospective simulation, enabling researchers to assess whether a policy maintains its efficacy over time. By comparing long-term trends across different policies, SYNAPSE facilitates the identification of stimulation strategies that remain robust and beneficial over extended periods.

**Zero-Shot Transfer Learning.**  Patient heterogeneity and disease variations in PD presents a major obstacle for generalization of RL policies. SYNAPSE enables the study of *zero-shot transfer*, where a policy trained on one virtual cohort is expected to perform reasonably well on another without significant fine-tuning. By evaluating performance drop-offs and adaptation requirements, this task allows SYNAPSE users to investigate robustness of RL agents to distribution shifts. Furthermore, it supports experimentation with domain generalization, meta-RL, and ensemble learning approaches to improve cross-patient adaptability. In general, SYNAPSE's virtual-cohort-based structure enables principled analysis of policy transferability, facilitating the development of broadly deployable aDBS strategies.

## 5  EVALUATION OF SYNAPSE

We evaluate the fidelity of our simulator from multiple aspects: (*i*) To evaluate the accurate modeling of historical data into a virtual cohort, we investigate embedding preservation via t-SNE visualizations, statistical alignment via rank correlation and mean absolute error (MAE) on the policies that have been applied to collect data using cross-validations. (*ii*) To evaluate the simulated trajectories under a new policy, we train our simulator using historical data collected from the set of the deployed policies except a hold-out policy, and roll-out trajectories under the hold-out policy to compare with actual trajectories under that policy. We investigate the per-step Earth Mover's distance

Table 3: Fidelity of Simulator on Embedding and Reward Prediction. (1) Variance explained by PCA (2D) and t-SNE KL divergence per virtual cohort across diverse policies. (2) Reward estimation consistency across different policies. We report rank correlation (Spearman) and mean absolute error (MAE) of evaluation policies between simulated rollouts vs. real logs (5,000 iterations). Standard deviations are .00 and omitted in the table.

| Virtual Cohort # | *1* | *2* | *3* | *4* | *5* |
|---|---|---|---|---|---|
| Variance Explained | 97% | 95% | 95% | 86% | 97% |
| KL divergence | 1.10 | 1.27 | .97 | 1.41 | 1.14 |
| Rank Corr. | 0.80 | 0.94 | 0.90 | 0.85 | 0.73 |
| MAE | 0.08 | 0.04 | 0.05 | 0.05 | 0.15 |

(EMD) between predicted and actual low-level step-wise rewards, and absolute error (AE) (scaled to $[0, 1]$) between predicted and actual high-level end-of-episode rewards. Moreover, we evaluate the per-step L2 distance for the predicted and actual states from simulated and actual trajectories, and the per-step L2 distance between actual states from two separate trajectories, under operation of the hold-out policy.

Table 3 reports metrics on embedding fidelity and reward prediction. The first two PCA components of our latent model capture $\approx$95–97% of the variance for most virtual cohorts (and 86% in Virtual Cohort# 4). The t-SNE KL divergences ($\approx$0.97–1.41) are also low, indicating that the simulated latent trajectories closely preserve the local and global structure of real patient trajectories. In terms of policy outcomes, the simulator's predicted cumulative returns are mostly highly correlated with the true returns (Spearman $\rho \approx$0.73–0.94 across virtual cohorts). The mean absolute error (MAE) of these return predictions is relatively small, especially for Virtual Cohorts# 1-4, meaning that not only the ranking but also the scale of policy performance is accurately reproduced.

As offline data is usually limited, and online test/trial budget may also be restricted, a testbed, that can support validate the possible reactions from patients and future outcome given a new treatment

Table 4: Off-policy Prediction and Evaluation. To investigate quality of simulated rewards, we report low-level per-step reward divergence (EMD) and high-level value error (AE) scaled on actual returns, for each held-out policy cohort. To investigate quality of simulated transitions, we report per-transition actual distance using the pairs of transitions, and per-transition sim distance using the pairs of transitions from simulated and true data, collected under the hold-out policy that is unseen during simulator training. Results are obtained from 100 runs.

| Virtual Cohort # | 1 | 2 | 3 | 4 | 5 |
|---|---|---|---|---|---|
| (Low-level reward) Per-step EMD | .32 (.17) | .26 (.13) | .17 (.08) | .31 (.06) | .22 (.04) |
| (High-level reward) AE | .36 (.05) | .22 (.08) | .17 (.18) | .08 (.09) | .9 (.03) |
| Per-transition true distance | 5.44 (1.41) | 1.94 (1.05) | .68 (.12) | 2.38 (.72) | 11.96 (7.4) |
| Per-transition sim-true distance | 18.34 (1.98) | 1.34 (0.74) | .51 (.1) | 3.52 (.49) | 11.54 (6.9) |

policy, is in need. Table 4 presents off-policy prediction and evaluation results for unseen patient records, *i.e.*, both interactions and the hold-out policy are unknown during training process. At the high level, for Virtual Cohorts# 3 & 4, the simulator achieves very low scaled absolute error in predicting the long-term return for most virtual cohorts, indicating near-optimal agreement, while Virtual Cohorts# 1 & 5 show larger errors, as opposed to the small errors they achieve when state-action coverage is observable in 3. This further indicates the highly complex nature of human-centered interactions. At the per-step level, we compute the Earth Mover's distance between the per-step reward distributions of real and simulated trajectories. These EMD values range roughly from 0.17 to 0.32 across virtual cohorts, confirming that the simulator reproduces the sequence of rewards (and thus behavior) very closely. In other words, both the aggregate returns and the time-series of intermediate rewards in simulation align well with the actual patient data (as reflected by the low AE and low EMD).

We also directly compared simulator rollouts to the real-patient trajectories as in Table 4. Especially for virtual cohorts# 2-5, we can observe the simulated-to-true per-transition distance is close to the real per-transition distance. Thus, the simulator may synthesize close-to-realistic interactions, even under a new policy that is never used before, indicating the potential benefits by using SYNAPSE as a testbed to validate new treatment strategies before deployment. In summary, simulated trajectories trace the similar trends as the recorded data, and summary statistics (e.g. mean return, variance) closely agree. These comparisons confirm that the simulator faithfully reproduces the behavioral data and reward statistics observed in the actual PD patients.

Despite the overall accuracy, there is some cohort-level variability in errors. For example, Virtual Cohort# 1 exhibits has a higher return error (high-level reward AE=0.36) and a notably larger per-transition deviation in simulation than other virtual cohorts (18.34 per-transition sim-true distance). Virtual Cohort# 4 also stands out with 86% variance explained in 2D (versus ≈95–97% for others) and higher high-level reward AE, indicating more complex or noisy dynamics under various policies. These residual errors likely stem from patient-specific factors. Parkinson's patients are heterogeneous in symptom profiles and responses to DBS, and session-to-session variability (e.g. medication effects or noise) can be substantial. Also, the fine-grained per-step rewards is not necessarily highly correlated with long-term end-of-episode rewards, indicating the challenges to explore the trade-off between prediction effects on short- and long-term outcomes of patients and better alignment between short- and long-term needs. In practice we observe that virtual cohorts with higher outcome variance tend to incur larger modeling errors: the virtual cohorts that are harder to predict (larger spread in their true returns) generally show higher EMD or AE. Such inter-patient variability constrains simulator accuracy, but even in these challenging cases the errors remain moderate.

**Leave-one-patient-out analysis.**   To further assess the fidelity of SYNAPSE on a held-out virtual cohort, we perform a leave-one-patient-out evaluation, where a mixed model is trained on data from all patients except one, which is reserved for testing. This setting is inherently more challenging due to population shift and the unique characteristics of each patient. We evaluate by comparing simulated rollouts with the true trajectories using multiple metrics: rank correlation and MAE of evaluation policies performance for hold-out patient; Earth Mover's Distance (EMD) for per-step reward divergence; absolute value error (AE) scaled by the actual returns; transition-level actual distance, computed from pairs of true transitions; and transition-level simulation distance, computed between simulated and true transitions. Table 5 summarizes the results of the leave-one-patient-out analysis.

Table 5: Off-policy Prediction and Evaluation for Leave-one-patient-out. We report rank correlation and MAE of evaluation policies performance for hold-out patient, low-level per-step reward divergence (EMD) and high-level value error (AE) scaled on actual returns, for each held-out patient. To investigate quality of simulated transitions, we report per-transition actual distance using the pairs of transitions, and per-transition sim distance using the pairs of transitions from simulated and true data, collected under the hold-out patient that is unseen during simulator training. Results are obtained from 100 runs.

| Hold-out Virtual Cohort # | 1 | 2 | 3 | 4 | 5 |
|---|---|---|---|---|---|
| Rank Corr. | 0.5 | 0.54 | 0.49 | 0.37 | 0.2 |
| MAE | 0.86 | 0.82 | 0.8 | 0.85 | 0.88 |
| (Low-level reward) Per-step EMD | 0.3 (0.08) | 0.37 (0.02) | 0.24 (0.08) | 0.35 (0.04) | 0.2 (0.08) |
| (High-level reward) AE | 0.3 (0.08) | 0.38 (0.01) | 0.26 (0.08) | 0.35 (0.03) | 0.2 (0.07) |
| Per-transition true distance | 5.55 (0.56) | 2.02 (0.51) | 1.09 (0.57) | 3.41 (0.68) | 15.19 (1.91) |
| Per-transition sim-true distance | 18.73 (0.94) | 2.12 (0.29) | 0.92 (0.25) | 11.24 (0.43) | 11.59 (1.93) |

Table 6: Analysis on Safety Constraints in Leave-one-patient-out. We report percentage (mean (std)) of transitions within 15% of max amplitude and within average value of true value +2SD in the target cohort of the rollout trajectories from simulation.

| Virtual Cohort | Within 15% Amp. | Within Avg.+2SD Energy |
|---|---|---|
| 1 | 0.98 (.02) | 0.50 (0.50) |
| 2 | 1.00 (.00) | 0.75 (0.43) |
| 3 | 0.95 (.05) | 0.50 (0.50) |
| 4 | 1.00 (.00) | 1.00 (.00) |
| 5 | 0.773 (.14) | 0.50 (0.50) |

Table 7: Analysis on Safety Constraints in Leave-one-policy-out within each virtual cohort. We report percentage (mean (std)) of transitions within 15% of max amplitude and within average value of true value +2SD in the target cohort of the rollout trajectories from simulation.

| Virtual Cohort | Within 15% Amp. | Within Avg.+2SD Energy |
|---|---|---|
| 1 | 0.996 (.06) | 1.00 (.00) |
| 2 | 1.00 (.00) | 1.00 (.00) |
| 3 | 1.00 (.00) | 1.00 (.00) |
| 4 | 1.00 (.00) | 1.00 (.00) |
| 5 | 0.842 (.36) | 1.00 (.00) |

**Analysis on Safety Constraints.** As discussed above, our simulator already embeds two safety constraints: (*i*) the action is defined as a percentage of the clinician-specified maximum amplitude for each participant, ensuring the policy cannot exceed the clinically permitted stimulation bounds, and (*ii*) each state–action update corresponds to a 2-second control cycle, preventing sub-second amplitude fluctuations that violate DBS programming guidelines. To further quantify safety, we perform a numerical analysis of the rollout trajectories from simulator by evaluating: (*i*) step-to-step amplitude changes (ramping) ($max|A_t - A_{t-1}| = 0.15$), i.e., $\leq 15\%$ of max amplitude per 2s, which corresponds to a full amplitude sweep in 13s and comfortably slower than the fastest constraints (250ms) reported in (Stanslaski et al., 2024), (*ii*) a normalized energy proxy proportional to the squared stimulation amplitude, with a threshold of average value of true value +2SD in the target cohort (Note that the energy usage of the learnt policies would not surpass that of cDBS because of the action constraints). We report the fraction of episodes that satisfy each constraint, in both leave-one-patient-out and leave-one-policy-out within a virtual cohort experiment, as in Tables 6 & 7.

## 6 CONCLUSIONS, LIMITATIONS, AND FUTURE WORKS

We introduced SYNAPSE, an open-source aDBS simulator for Parkinson's disease, which serves as a realistic testbed for both offline, online RL, and outcome prediction tasks in neurostimulation, supporting policy development and evaluation. This platform enables the design and evaluation of adaptive DBS policies through close alignment with real patient data and clinical constraints. The training data come from a limited number of patients, reflecting the challenges of clinical data collection. Nonetheless, each patient contributed rich longitudinal records spanning multiple years, while we are the first work to open-sourced such simulator to related community for initiating new algorithms. Additionally, while the simulator already mirrors patient-specific dynamics with high accuracy, future enhancements could explore more expressive modeling techniques to further improve realism. All such advancements must remain mindful of safety and privacy constraints given the sensitive nature of patient data. Future work will expand the simulator to larger, more diverse patient cohorts and incorporate additional sensor streams and medication schedules for richer context. We also plan broader dissemination of our platform, aiming for it to become a standard resource in human-centric RL research and to catalyze further innovations in data-driven neuromodulation. Moreover, incorporating a broader OPE/offline RL evaluation suite (e.g. EpiCare (Hargrave et al., 2024)) would be an interesting direction to foster utility of SYNAPSE. Indeed, SYNAPSE provides a platform that is generalizable to OPE/RL algorithms; we are open to any future collaboration opportunities to facilitate advanced techniques in both RL and DBS.

ETHICS STATEMENT

This research involves the development of a simulator for adaptive deep brain stimulation (aDBS) in Parkinson's disease based on longitudinal clinical data. All data used in this work were collected under Institutional Review Board (IRB) approval, with informed consent obtained from all participants. The simulator is constructed using de-identified records that comply with HIPAA and institutional data-sharing policies. Access to the original patient data is restricted and requires appropriate institutional affiliation and approval.

Our simulator does not attempt to replace clinical decision-making, but instead serves as a research tool for evaluating machine learning or reinforcement learning algorithms in a safe, reproducible, and privacy-preserving environment. To safeguard patient privacy and prevent data leakage, we do not release raw clinical data at this beta version. The open-source release includes only learned models and policy evaluation tools, accompanied by usage documentation that enforces responsible use. We acknowledge the small participant size as a limitation and take care not to over-generalize findings. Fairness and bias mitigation are considered through individualized cohort modeling to respect inter-patient heterogeneity. The simulator explicitly encodes safety constraints from real-world hardware and clinical practice to prevent unsafe actions during policy learning.

REPRODUCIBILITY STATEMENT

Complete code and pretrained model checkpoints will be released publicly upon acceptance of the paper. To preserve the double-blind review process, given the sensitivity of the data and checkpoints, we may defer public release until after acceptance. Upon publication, we will provide a permanent, citable repository with safeguard and maintain detailed documentation to support future researchers in reproducing and extending our findings.

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

## A    MORE DISCUSSIONS ON THE ADBS FOR PD PARTICIPANT SIZE

Parkinson's aDBS trials are inherently small due to their complexity. Each "complete" trajectory requires months of testing and careful clinical oversight. For example, Gao et al. (2023c) analyzed 4 patients (our 5 is comparable) and other aDBS pilot studies often involve ¡10 subjects (Little et al., 2013; 2016a; Rosa et al., 2015; Malekmohammadi et al., 2016; Swann et al., 2018). Thus, 5 patients is not atypical for this domain. Importantly, each patient in our dataset underwent multiple stimulation policies (standard cDBS, no stimulation, various experimental strategies) over many sessions, yielding rich longitudinal data. In total we have on the order of thousands of transitions across cohorts. We will clarify this context to show how even large trials require substantial resources. While we fully agree that more patients would further improve generality, our current 5-cohort dataset (collected over 3 years) is in line with the scope of realistic DBS experimentation, and it already captures significant inter-individual variability through personalized modeling. We will highlight these points in the revision, and note that SYNAPSE's framework can incorporate additional patients as data become available.

## B    MORE DISCUSSIONS ON UNDERSTANDING CLINICALLY OUTCOMES USING SYNAPSE

In our setting, the episodic return is directly tied to clinically relevant signals. Specifically, our per-step reward includes reductions in pathologic LFP beta-band power, and clinical research shows that high-beta LFP power correlates with motor symptoms: *e.g.*, higher subthalamic high-$\beta$ power is strongly associated with greater improvement in bradykinesia–rigidity (Chen et al., 2022). In practical terms, maximizing our simulated return (*i.e.*, minimizing LFP power over time) corresponds to reducing a patient's Parkinsonian symptoms. Moreover, the human feedback (HF) we include is based on patient-level outcomes (similar to patient-reported satisfaction). Thus, policies that achieve higher episode return in SYNAPSE are those expected to yield better symptom relief and patient feedback in real therapy. In future work we plan to quantify this mapping explicitly (*e.g.*, regress UPDRS scores vs. model returns), but for now we emphasize that our reward design is grounded in known biomarkers.

## C    DISCUSSIONS ON NEW PATIENT ARRIVAL USING SYNAPSE

A nature question for future empirical trials with SYNAPSE could be how one would choose which virtual cohort to test a new policy for a given patient. One could imagine strategies such as clustering patients by baseline features (*e.g.*, symptom profile, baseline beta power statistics) or by distance of initial response trajectories. For instance, given a new patient's early DBS data, we could compute distances to each virtual cohort's trajectories to identify the closest match. In practice, one might also "cold-start" with a safe generic policy (*e.g.*, a clinician-tuned controller) and then personalize it based on pre-approved policy candidates, where reliable off-policy evaluation approaches are necessary to perform such challenging tasks.

## D    DETAILED ENVIRONMENT SETUP

### D.1    ADAPTIVE NEUROSTIMULATION FOR PARKINSON'S DISEASE.

Adaptive neurostimulation has shown promise in treating a range of neurological conditions (Benabid, 2003; Deuschl et al., 2006; Follett et al., 2010; Okun, 2012b). Deep Brain Stimulation (DBS), in particular, is commonly used for managing Parkinson's disease (PD). This approach involves surgically implanting an internal pulse generator (IPG) beneath the collarbone, which delivers electrical pulses to the basal ganglia (BG) via electrodes inserted into the brain. An overview of the system configuration is presented in Figure 4.

Adaptive DBS (aDBS) enhances this framework by modulating the stimulation amplitude in real time, responding dynamically to irregular neural activity associated with PD. This modulation is driven by real-time feedback from local field potentials (LFPs), which serve as physiological indicators—analogous to environmental rewards in a reinforcement learning (RL) framework.

Several studies have applied RL to simulate aDBS strategies using computational models of the BG (Guez et al., 2008; Gao et al., 2020; Nagaraj et al., 2017; Pineau et al., 2009). In these models, the reward signal is often defined using the beta-band power spectral density of LFPs—referred to as beta power—which tends to increase in PD due to abnormal neural synchronization (Kuncel & Grill, 2004).

However, in clinical settings, patient-reported outcomes do not always align consistently with beta power fluctuations. This inconsistency stems from the heterogeneity of PD symptoms and progression across individuals Okun (2012a); Kühn et al. (2006); Brown et al. (2001); Wong et al. (2022). These observations underscore the importance of evaluating human feedback (HF) or human-centered returns directly in real-world deployments, highlighting the relevance of RL with human feedback (RLHF) for safe and personalized neuromodulation.

Figure 4: Setup of the neurostimulation experiments, as well as the formulation of offline trajectories. Environmental rewards and human returns are captured in streams 1 and 2-3 respectively.

## D.2 DBS Hardware and Implementation

The Summit RC+S system includes an implantable pulse generator (IPG) that supports both neuromodulation and real-time sensing. It also provides software APIs to interface with the system, enabling implementation of reinforcement learning (RL) policies for clinical testing. The full control pipeline is shown in Figure 5. Each MDP step spans 2 seconds: during this window, the IPG records local field potentials (LFPs) from the basal ganglia (BG) at 500 Hz. These signals are transmitted wirelessly to a research tablet, where they undergo processing to extract beta-band power spectral densities (beta power), which form the MDP state and reward.

Beta power is calculated using a Fast Fourier Transform (FFT) with 512-bin resolution over the 2-second LFP window. While the beta-band typically spans 13–35 Hz, its exact range was

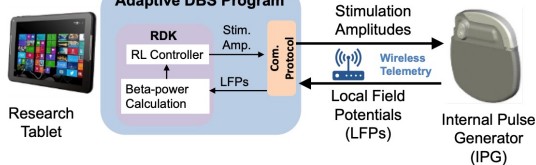

Figure 5: Implementation of RL control on the research tablet over the API provided by the manufacturer of the DBS device. During clinical testing, the research tablet evaluates the RL policies which determine the stimulation amplitudes to be used in each step (every 2 seconds). Then, by communicating with the internal pulse generator (IPG) through wireless telemetry, the IPG adjust the stimulus accordingly and sends back the local field potentials (LFPs) which are used to determine the MDP state and reward for the next step.

customized per patient to better reflect abnormal neural oscillations; details are provided in the Patient Characteristics section. Based on the most recent beta power estimates, the RL policy selects a stimulation amplitude for the next cycle, which is sent wirelessly to the IPG. Electrodes with four contacts were implanted in each hemisphere targeting the subthalamic nucleus (STN) and globus pallidus (GP). Stimulation was delivered in monopolar mode via a single contact, with the IPG case serving as the return electrode. The two contacts surrounding the stimulation contact were used for sensing LFPs (i.e., sandwich sensing).

The IPG is FDA-approved for research, and software modifications to enable adaptive RL-based stimulation were cleared under an Investigational Device Exemption (IDE). All procedures were approved by the IRB, and participants were compensated in line with institutional policies.

## D.3 RL Policies Tested

Target RL policies are all trained using DDPG. Specifically, each is trained over a growing experience buffer containing the data collected from the latest trials, in addition to the fixed set of offline trajectories above; this would lead to a set of target policies with varied performance, as the policies

obtained in the later stage tend to be more optimal since the environment has been explored to a broader extent, compared to the policies obtained at earlier stages. Moreover, the policies have to demonstrate at least moderate control efficacy (as determined by clinicians), in order to be considered as target policies; since the ground-truth return of each target policy is determined by averaging the human returns over more than 10 sessions of clinical testing ($> 100$ minutes in total). Note that, for each patient, there may exist a few policies that lead to close returns, due to the testing limitations introduced above, *i.e.*, the use of $\xi$ as well as the safety protocol for long-term testing (introduced above).

Every 2 seconds, the action is the chosen stimulation amplitude (a percentage of the device's maximum output) within a patient-specific safe range (for example, $\xi = 40\text{–}60\%$ as a lower bound). The data is collected in both at-home care and clinical visits, and each episode can be operated under a policy. A patient can have experienced multiple treatment policies (depending on their health conditions evaluated by clinicians), including two main types: traditional controllers and RL policies. Early in data collection, two simple controllers were used: a uniform random controller (sampling amplitudes uniformly from $[\xi, 100\%]$) and a clinician-tuned proportional–integral (PI) controller. They help identify the state-action regions that can lead to non-trivial returns. Then, such data are used to train RL policies in a prior research – following DDPG. Overseen by clinicians and related departments, only a small number of RL policies were actually tested on each patient.

### D.4    MORE DETAILED ENVIRONMENTAL SETUP FOR SYNAPSE

Each discrete step in HMDP corresponds to a control cycle which lasts 2 seconds. Each episode lasts at least $T = 300$ steps (*i.e.*, the horizon), or more than 10 minutes in the real world. Specifically, states are constituted by a historical sequence of beta powers obtained from past 10 steps, following a similar setup as in (Gao et al., 2020; Nagaraj et al., 2017). The actions $a \in \mathcal{A}$ are in $[\xi, 100\%] \subset \mathbb{R}$, $0\% \leq \xi < 100\%$, which represent the percentage over the maximum amplitude of the stimulus the IPG can employ. Specifically, $\xi = 40\%$ for patient #0 and $\xi = 60\%$ for patients #1-#4, to ensure safety and minimum efficacy of DBS during clinical testing. The environmental rewards are defined following, *i.e.*,

$$R(s, a, s') = \begin{cases} -1 - 0.2a & \text{if latest beta power is above a threshold,} \\ -0.2a & \text{if latest beta power is below a threshold;} \end{cases} \tag{1}$$

here, $-0.2a$ is the penalty for stimulating with higher-than-needed amplitudes, to preserve the runtime of the IPG device (powered by a rechargeable battery) between recharges, and an additional $-1$ is given if stimulating with the amplitude determined by the policy cannot reduce beta power to below a threshold specific to each patient. The thresholds above are set to be the lower 20% quantile of the beta powers observed from initial exploration (introduced below). The human return at the end of each episode is quantified as a 66.7%–33.3% weighted sum over the hand grasp speed captured from performing the maneuver (rapid and full extension and close of all fingers) for the bradykinesia test, as well as the proportional length of the session where the patient displays tremor (as captured by a wearable accelerometry).

## E    MORE DETAILED METHODOLOGY OF BUILDING SYNAPSE

We build a generative model, which enables high-fidelity trajectory simulation and off-policy evaluation. In general, we use an off-the-shelf algorithm, the variational latent model with human returns (VLM-H) (Gao et al., 2023b), that has achieved desirable and robust performance in related offline scenarios to capture the environmental dynamics in aDBS. We train VLM-H on a corpus of historical data collected from various policy interactions from real aDBS experiments.

To model offline state-action sequences for reinforcement learning and off-policy evaluation (OPE), we build upon the VLM-H originally proposed in Gao et al. (2023a). Unlike earlier VAE-based methods that focus on improving sample efficiency for policy learning Lee et al. (2020); Zhang et al. (2019); Rybkin et al. (2021); Hafner et al. (2019; 2020a;b), VLM was designed to facilitate OPE by learning a compact and expressive latent space where state-action pairs are organized according to the return of their originating policies. To incorporate additional human feedback, we use VLM-H, which extends the VLM by integrating human returns $G_{0:T}^{\mathcal{H}}$ into the model. Specifically, VLM-H

includes a latent prior $p(z)$ over variables $z \in \mathbb{R}^L$, an encoder $q_\psi(z_t|z_{t-1}, a_{t-1}, s_t)$, and two decoders: $p_\phi(z_t, s_t, r_{t-1}|z_{t-1}, a_{t-1})$ for per-step reconstruction of transitions, and $p_\phi(G_{0:T}^{\mathcal{H}}|z_T)$ for episode-level human return prediction. All distributions are parameterized as diagonal Gaussians with neural networks $\phi$ and $\psi$. The model is trained by maximizing an evidence lower bound (ELBO) that jointly reconstructs states, environmental rewards, and human returns, while regularizing the latent space via KL divergence terms. This design allows VLM-H to capture both dynamics and evaluative signals relevant to downstream policy learning and evaluation.

To train $\phi$ and $\psi$, we maximize the evidence lower bound (ELBO) of the joint log-likelihood $\log p_\phi(s_{0:T}, r_{0:T-1}, G_{0:T}^{\mathcal{H}}|\phi, \psi, \rho^\beta)$, *i.e.*,

$$
\max_{\psi,\phi} \quad \mathbb{E}_{q_\psi}\Big[ \log p_\phi(G_{0:T}^{\mathcal{H}}|z_T) + \sum\nolimits_{t=0}^{T} \log p_\phi(s_t|z_t) + \sum\nolimits_{t=1}^{T} \log p_\phi(r_{t-1}|z_t)
$$

$$
- KL\big(q_\psi(z_0|s_0)||p(z_0)\big) - \sum\nolimits_{t=1}^{T} KL\big(q_\psi(z_t|z_{t-1}, a_{t-1}, s_t)||p_\phi(z_t|z_{t-1}, a_{t-1})\big)\Big]; \quad (2)
$$

*Model Selection.* To achieve more realistic performance in simulating real patient's interactions, we select models from trained generative models with best estimated values in terms of both rank correlation and mean absolute error (MAE), which are two standard evaluation metrics in OPE, calculated from estimated and true policy values under target policies. Specifically, we select the models with the highest rank correlation and then lowest MAE if rank correlation is in a tie.

*Model Architecture & Hyper-parameters* In the VLM-H framework, both the encoder $q_\psi(z_t|z_{t-1}, a_{t-1}, s_t)$ and the decoder $p_\phi(z_t|z_{t-1}, a_{t-1})$ are implemented as LSTMs with 64 hidden units, each followed by two fully connected layers containing 128 and 64 neurons, respectively. The remaining components of the encoder–decoder architecture—namely $q_\psi(z_0|s_0)$, $p_\phi(G_{0:T}^{\mathcal{H}}|z_T)$, $p_\phi(s_t|z_t)$, and $p_\phi(r_{t-1}|z_t)$—are realized as multi-layer perceptrons (MLPs), each composed of two dense layers with 128 and 64 nodes. Following the approach in Gao et al. (2023a), we incorporate a regularization term into the ELBO equation 2, scaled by a coefficient $C'$ to balance the magnitude of the two ELBO components; this regularizer is applied to the LSTM latent states between $q_\psi(z_t|z_{t-1}, a_{t-1}, s_t)$ and $p_\phi(z_t|z_{t-1}, a_{t-1})$. The scaling factor $C'$ is selected from the set $\{.01, .1, .05, 1., 10\}$. The learning rate is chosen via grid search over $\{0.001, 0.0007, 0.0003, 0.0001\}$ and is subject to an exponential decay factor of 0.997 applied at every iteration. Training is carried out for 20 epochs with a mini-batch size of 64 using the Adam optimizer. We apply $L_2$ weight decay with a coefficient of 0.001 and batch normalization to all hidden fully connected layers. Finally, the ten checkpoints achieving the highest ELBO scores are forwarded to the RILR stage.

# F    INTERACTIVE USE CASES IN SYNAPSE

Here, we exhibit a user case on a general purpose of learning RL policies and testing patient's outcome given a policy, by interacting with SYNAPSE. Note this is providing an example of a set of tasks, but not covering all functions and tasks provided by SYNAPSE.

## F.1    OFFLINE POLICY TRAINING AND OUTCOME PREDICTION

To perform offline policy learning, a user can use a pre-trained CQL (Kumar et al., 2020) agent on logged trajectories from a given virtual cohort (e.g., Virtual Cohort #4). For instance, using an offline-RL library like `d3rlpy`, one would load the stored dataset of (state, action, reward) transitions and configure a CQL agent. In `d3rlpy`, this may look like creating a `CQL` config and fitting it on the dataset. The key hyperparameters for training were set as follows:

- Optimizer: Adam for both actor and critic
- Learning rates: $3 \times 10^{-4}$ for both actor and critic
- Other settings: as in our implementation (e.g., batch size, number of training steps)

After offline training, the policy is evaluated by deploying it back into the SYNAPSE simulator. The user ran the trained agent on the Virtual Cohort# 4 environment to generate new rollouts. The

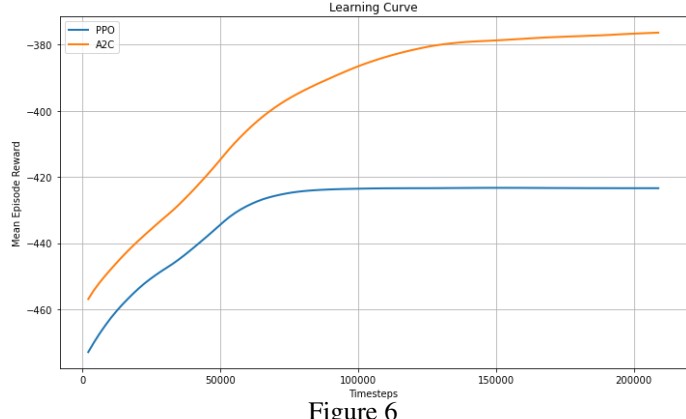

Figure 6

observed average return (direct sum over per-step rewards) over these trajectories was approximately -415.27 ± 0.73. This negative return is expected in our setting: per-step rewards in SYNAPSE are generally negative due to the Parkinson's-specific reward shaping, so even a good policy can have a negative cumulative reward.

## F.2 ONLINE POLICY TRAINING

We also demonstrate on-policy RL by training PPO (Schulman et al., 2017) and A2C (Mnih et al., 2016) agents in the SYNAPSE environment (via the `GymLearnedEnv` wrapper). Both agents use `MlpPolicy` networks and log training metrics to TensorBoard for monitoring. The example hyperparameters were:

- A2C: learning rate $3 \times 10^{-4}$, discount $\gamma = 0.95$, GAE $\lambda = 0.9$, $n_{\text{steps}} = 2048$
- PPO: learning rate $3 \times 10^{-4}$, batch size 64, discount $\gamma = 0.95$, GAE $\lambda = 0.9$, clip range 0.1

The user trained each model for a fixed number of episodes and recorded the learning curves. The results are shown in Figure 6.

## F.3 REPRODUCIBILITY AND MODULARITY OF SYNAPSE

These case studies illustrate SYNAPSE's flexibility as a unified RL testbed. Users can seamlessly switch workflows using the same patient-specific environment model. For example, SYNAPSE enables offline policy learning from logged data *and* online real-time policy evaluation. By supporting standard RL interfaces (Gym API) and detailed logging, SYNAPSE makes experiments fully reproducible and modular. In summary, SYNAPSE is an open-source simulator that serves as a realistic testbed for both offline and online RL and time-series analysis in adaptive neurostimulation. Its patient-specific cohort models and common API allow researchers to develop and compare algorithms across modalities (offline learning, on-policy training, OPE, etc.).

