# OpenReview forum: "SYNAPSE: Simulation Benchmark of Neuro-Adaptive Patient-Specific Evaluation for Episodic Decision-Making"
_ICLR.cc/2026/Conference — Submitted to ICLR 2026_

### Official Review · Reviewer_5ezG · 2025-10-25

**Soundness:** 2
**Presentation:** 3
**Contribution:** 3
**Rating:** 6
**Confidence:** 4

**Summary:**

This paper develops a simulation for aDBS for Parkinsons. The approach uses clinical data to train models of human patients. It enables future aDBS policy to be tested in silico before deployment on real patients.

**Strengths:**

aDBS is a very promising technology to improve the lives of patients. However, there do not currently exist many ways to effectively test new methods for aDBS control before deployment in the patient. This paper addresses this very important problem by using patient derived data  to create an aDBS simulation.

**Weaknesses:**

Immediate rewards and ihr could be better explained when initially introduced.

More details about the actual modeling need to be included since this is the point of the paper. This shouldnt just be in the appendix. A diagram of the simulation architecture would also be helpful

A discussion of how one would choose which environment to test a new policy for a given patient would be helpful. is there a way to quantify how close a new patient is to each simulation environment to know which would be best to use?

More detail about the heterogeneity of responses would be useful and differences in each environment would be useful

Related works section talking about other data-driven human simulations for medical applications would be useful to better contextualize this work

This is an unusual topic for ICLR. I would recommend briefly discussing "Coprocessor Actor Critic: A Model-Based Reinforcement Learning Approach For Adaptive Brain Stimulation"  in related works as this is the most similar work that has also been published in this community

**Questions:**

New control strategies be out of distribution of the policies that the simulation was trained on. Is there a way to quantify uncertainty in the simulation?
What does turning off dbs mean? Is this just no stimulation?
Can medication be modeled in the simulation?

---

> ### Author Response · Authors · 2025-11-23
> **Rebuttal by Authors (1/N)**
>
> We sincerely appreciate your time and efforts on evaluating our work, and your positive comments that the paper is making an important empirical impact. Please find our point-by-point response below.
>
> >Q1. Immediate rewards and ihr could be better explained when initially introduced.
>
> A1. We thank the reviewer for noting this and will make the definitions explicit.In our human-involved MDP (HMDP), immediate reward and immediate human reward (IHR) are distinct signals. Specifically, the immediate reward $R(s,a)$ is a deterministic function combining the change in beta-band power and stimulation amplitude. In contrast, the immediate human reward (IHR) is a stochastic feedback signal drawn from a learned distribution $R^H$ at each time step. This IHR captures short-term patient responses (e.g. tremor/bradykinesia outcomes) that are not captured by beta power alone. **We’ve revised our description in the Section 4, Environmental Setup (highlighted in blue)**.
>
> >Q2. More details about the actual modeling need to be included since this is the point of the paper. This shouldnt just be in the appendix. A diagram of the simulation architecture would also be helpful
>
> A2. We agree that the core modeling components deserve clearer exposition in the main paper. Due to page limits, we initially placed full technical details in Appendix, but **we’ve included a new schematic diagram (Figure 2)** of the simulator architecture.
>
> >Q3. A discussion of how one would choose which environment to test a new policy for a given patient would be helpful. is there a way to quantify how close a new patient is to each simulation environment to know which would be best to use?
>
> A3. Great suggestion. One could imagine strategies such as clustering patients by baseline features (e.g., symptom profile, baseline beta power statistics) or by distance of initial response trajectories. For instance, given a new patient’s early DBS data, we could compute distances to each virtual cohort’s trajectories to identify the closest match. In practice, one might also “cold-start” with a safe generic policy (e.g., a clinician-tuned controller) and then personalize it based on pre-approved policy candidates, where reliable off-policy evaluation approaches are necessary to perform such challenging tasks. We will add a discussion noting that selecting the most appropriate virtual cohort for a new patient is a challenging open problem in off-policy evaluation, and suggest some approaches. **We’ve added such discussion to Appendix C (highlighted in blue).**
>
> >Q4. More detail about the heterogeneity of responses would be useful and differences in each environment would be useful
>
> A4. We agree that highlighting patient heterogeneity of responses is valuable. We’ve added more descriptive details of responses (in terms of human rewards) to Table 2 (highlighted in blue).
>
> >Q5. Related works section talking about other data-driven human simulations for medical applications would be useful to better contextualize this work
>
> A5. We appreciate the suggestion to broaden the related work. We will expand Section 2 to discuss other healthcare simulation/benchmark efforts. For example: GraphSim provides a general graph-based model trained on ICU/EHR data [1]. MediSim is a generative framework that simulates realistic patient trajectories by filling in missing visits and data modalities in EHRs [2]. This contextualizes our contribution as the first such patient-specific simulator for adaptive DBS, complementing digital twins in other domains.
>
> >Q6. This is an unusual topic for ICLR. I would recommend briefly discussing "Coprocessor Actor Critic: A Model-Based Reinforcement Learning Approach For Adaptive Brain Stimulation" in related works as this is the most similar work that has also been published in this community
>
> A6. We thank the reviewer for this reference. Pan et al. (2024) Coprocessor Actor Critic work presents a model-based RL approach for adaptive brain stimulation. We will explain that their focus is on the RL algorithm that learns an effective policy via an in silico model of brain injury, whereas our focus is on providing a realistic simulation benchmark (trained on real patient DBS data) for evaluating any such RL algorithms. In other words, their work and SYNAPSE are complementary: both address personalized neuromodulation with RL, but one contributes a new learning method and the other provides an environment to test methods. **We’ve cited and clarified our approach with Pan et al. noting our use of actual patient recordings versus their computational brain model in Introduction and Section 2.2 (highlighted in blue).**

---

> > ### Author Response · Authors · 2025-11-23
> > **Rebuttal by Authors (2/N)**
> >
> > >Q7. New control strategies be out of distribution of the policies that the simulation was trained on. Is there a way to quantify uncertainty in the simulation? What does turning off dbs mean? Is this just no stimulation? Can medication be modeled in the simulation?
> >
> > A7. We recognize that novel control strategies may fall outside the distribution of our training data. To partially address this, we evaluated the simulator on held-out policies (Tables 3 & 4).The results show that even for unseen policies, the simulated trajectories closely match real patient data. We further added evaluations on held-out cohorts (Tables 5 & 6, highlighted in blue), we see due to the heterogeneity across patients, the uncertainty increased for partial cohorts (e.g. cohort 2) in terms of simulated transitions and rewards.
> > In our experiments, turning off dbs means no stimulation (the IPG output is set to zero for that period). We’ve added clarifications in the paper (**Section 3 Clinical Sessions, highlighted in blue**) to avoid confusion (it is essentially the zero-amplitude control baseline).
> >
> > Our current simulator does not explicitly model medication adjustments. All patient data were collected under the patients’ prescribed Levodopa equivalent doses, so variations in medication are implicitly captured in the observed dynamics. Incorporating medication as an input variable is indeed a valuable extension and would require richer time-series (e.g. tracking dose timing). We’ve mentioned this as a future work: richer multi-modal data (medication schedules, more sensors) could enable a more comprehensive simulation of PD management (added in **Section 6, highlighted in blue**).
> >
> > We hope these answers provide some explanations to address your concerns and showcase that our work is solving a significant challenge in a satisfying manner. We are happy to answer any followup questions or hear any comments from you.
> >
> > References
> >
> > [1] Liventsev, Vadim, Aki Härmä, and Milan Petković. "Towards effective patient simulators." Frontiers in artificial intelligence 4 (2021): 798659.
> >
> > [2] Theodorou, Brandon, et al. "MediSim: Multi-granular simulation for enriching longitudinal, multi-modal electronic health records." Patterns (2025).

---

> ### Comment · Reviewer_5ezG · 2025-11-27
> **Response to rebuttal**
>
> Thank you to the authors for their thorough response. I believe all of my concerns have been addressed and i will raise my score to  a 7

---

> > ### Author Response · Authors · 2025-11-28
> > **Thank you!**
> >
> > Thank you very much for your thoughtful evaluation and for raising your score! We sincerely appreciate your careful reading and constructive feedback, which helped strengthen our paper. We understand that review editing may be currently disabled, but we are grateful for your supportive update and your acknowledgment that the concerns have been addressed.

---

### Official Review · Reviewer_zV95 · 2025-10-31

**Soundness:** 3
**Presentation:** 3
**Contribution:** 2
**Rating:** 4
**Confidence:** 3

**Summary:**

This paper develops a SImulation Benchmark of Neuro-Adaptive PatientSpecific Evaluation (SYNAPSE) for adaptive deep brain stimulation (DBS) in Parkinson’s disease. The paper includes offline training and evaluation of different treatment strategies, reflecting both short- and long-term effects.

**Strengths:**

1.	The developed simulator is critical to understand the impact of reinforcement learning on DBS and bridge the gap between evaluation and real-world clinical practice.
2.	The paper includes a diverse set of modern RL algorithms across multiple reward definitions and time horizons.

**Weaknesses:**

1.	The contribution of this paper needs clarification. The experiment and the developed dataset have significant overlap with the existing work [1]. The paper should explicitly articulate what the new components are provided in the newly proposed benchmark.
2.	The dataset includes only five patients, which limits the generality and makes the term “benchmark” somewhat premature. While the authors acknowledge this limitation, a stronger justification is needed to argue why this sample is sufficient to model variability across the PD population.
3.	While standard RL metrics (e.g., episodic return) are used, there is a lack of discussion on how these results map to clinically meaningful outcomes.
4.	Learning Agent is not defined in Figure 2. Does it refer to an RL agent?

[1] Gao, Qitong, Stephen L. Schmidt, Afsana Chowdhury, Guangyu Feng, Jennifer J. Peters, Katherine Genty, Warren M. Grill, Dennis A. Turner, and Miroslav Pajic. "Offline learning of closed-loop deep brain stimulation controllers for parkinson disease treatment." In Proceedings of the ACM/IEEE 14th International Conference on Cyber-Physical Systems (with CPS-IoT Week 2023), pp. 44-55. 2023.

**Questions:**

1.	What are the key differences between SYNAPSE and the previous dataset/work [1]?

---

> ### Author Response · Authors · 2025-11-23
> **Rebuttal by Authors (1/N)**
>
> Thank you for your time and efforts on evaluating our work, and your positive comments that our work is critical across domains. Please find our point-by-point response below.
>
> >Q1. The contribution of this paper needs clarification. The experiment and the developed dataset have significant overlap with the existing work [1]. The paper should explicitly articulate what the new components are provided in the newly proposed benchmark.
>
> A1. We appreciate the opportunity to highlight our paper’s novel contributions. Unlike Gao et al. [1], which proposed an offline RL pipeline for aDBS using a small patient dataset, SYNAPSE is a comprehensive simulation benchmark built from real patient data. Key distinctions include:
>
> &nbsp;&nbsp;&nbsp;&nbsp;(i) Full MDP simulator and benchmark: We construct a high-fidelity episodic MDP for aDBS, incorporating patient-specific dynamics and outcomes. Gao et al. focused on learning a policy directly from data, whereas we provide an environment that can be used to train and evaluate any decision-making method.
>
> &nbsp;&nbsp;&nbsp;&nbsp;(ii) Cohort-specific generative modeling: Our approach uses a variational latent model (VLM-H with a  regularization) to capture both short-term LFP dynamics and episode-level outcomes per cohort. This personalized simulation of each patient’s responses is a new feature; Gao et al. did not offer a reusable per-cohort simulator.
>
> &nbsp;&nbsp;&nbsp;&nbsp;(iii) Human feedback and long-term effects: We explicitly model the delay between per-step stimulation effects and cumulative patient feedback (IHRs vs. HF), reflecting real clinical evaluations. Gao et al. did not introduce this episodic HF framework or a separate long-horizon outcome predictor. (iv) Extensive evaluation support: We validate SYNAPSE’s realism against held-out data and provide baseline results under both standard cDBS and learned policies, offering clear guidance and infrastructure for future policy development and selection, whereas Gao et al. evaluated a specific controller.
>
> **A brief discussion of [1] has been added to Section 2.2 (highlighted in blue)**.
>
> >Q2. The dataset includes only five patients, which limits the generality and makes the term “benchmark” somewhat premature. While the authors acknowledge this limitation, a stronger justification is needed to argue why this sample is sufficient to model variability across the PD population.
>
> A2. Parkinson’s aDBS trials are inherently small due to their complexity. Each “complete” trajectory requires months of testing and careful clinical oversight. For example, Gao et al. [1] analyzed 4 patients (our 5 is comparable) and other aDBS pilot studies often involve <10 subjects [1-6]. Thus, 5 patients is not atypical for this domain. Importantly, each patient in our dataset underwent multiple stimulation policies (standard cDBS, no stimulation, various experimental strategies) over many sessions, yielding rich longitudinal data. In total we have on the order of thousands of transitions across cohorts. **We’ve added clarifications and discussions in this context (Appendix A (highlighted in blue))** to show how even large trials require substantial resources.
> While we fully agree that more patients would further improve generality, our current 5-cohort dataset (collected over 3 years) is in line with the scope of realistic DBS experimentation, and it already captures significant inter-individual variability through personalized modeling. We will highlight these points in the revision, and note that SYNAPSE’s framework can incorporate additional patients as data become available.
>
> >Q3. While standard RL metrics (e.g., episodic return) are used, there is a lack of discussion on how these results map to clinically meaningful outcomes.
>
> A3. Great question. In our setting, the episodic return is directly tied to clinically relevant signals. Specifically, our per-step reward includes reductions in pathologic LFP beta-band power, and clinical research shows that high-beta LFP power correlates with motor symptoms: e.g., higher subthalamic high-β power is strongly associated with greater improvement in bradykinesia–rigidity [7]. In practical terms, maximizing our simulated return (i.e. minimizing LFP power over time) corresponds to reducing a patient’s Parkinsonian symptoms. Moreover, the human feedback (HF) we include is based on patient-level outcomes (similar to patient-reported satisfaction). Thus, policies that achieve higher episode return in SYNAPSE are those expected to yield better symptom relief and patient feedback in real therapy. In future work we plan to quantify this mapping explicitly (e.g. regress UPDRS scores vs. model returns), but for now we emphasize that our reward design is grounded in known biomarkers. **We’ve expanded the discussion to make this connection clear and cite such clinical correlations in Appendix B (highlighted in blue)**.

---

> > ### Author Response · Authors · 2025-11-23
> > **Rebuttal by Authors (2/N)**
> >
> > >Q4. Learning Agent is not defined in Figure 2. Does it refer to an RL agent?
> >
> > A4. Thanks for catching this. We’ve updated the caption in Figure 2, “Learning Agent” refers to the RL agent/policy being trained to interact with the DBS simulator.
> >
> > We hope these answers address your questions and showcase that our work is solving a significant challenge in a satisfying manner. We are happy to answer any followup questions or hear any additional comments.
> >
> > References
> >
> > [1] Gao, Qitong, et al. "Offline learning of closed-loop deep brain stimulation controllers for parkinson disease treatment." In Proceedings of the ACM/IEEE 14th International Conference on Cyber-Physical Systems (with CPS-IoT Week 2023), pp. 44-55. 2023.
> >
> > [2] Little, Simon, et al. "Adaptive deep brain stimulation in advanced Parkinson disease." Annals of neurology 74.3 (2013): 449-457.
> >
> > [3] Little, Simon, et al. "Bilateral adaptive deep brain stimulation is effective in Parkinson's disease." Journal of Neurology, Neurosurgery & Psychiatry 87.7 (2016): 717-721.
> >
> > [4] Rosa, Manuela, et al. "Adaptive deep brain stimulation in a freely moving Parkinsonian patient." Movement Disorders 30.7 (2015): 1003.
> >
> > [5] Malekmohammadi, Mahsa, et al. "Kinematic Adaptive Deep Brain Stimulation for Resting Tremor in Parkinson’s Disease." Movement disorders 31.3 (2016): 426-428.
> >
> > [6] Swann, Nicole C., et al. "Adaptive deep brain stimulation for Parkinson’s disease using motor cortex sensing." Journal of neural engineering 15.4 (2018): 046006.
> >
> > [7] Chen, Po-Lin, et al. "Subthalamic high-beta oscillation informs the outcome of deep brain stimulation in patients with Parkinson's disease." Frontiers in Human Neuroscience 16 (2022): 958521.

---

> > > ### Comment · Reviewer_zV95 · 2025-11-26
> > >
> > > The response has addressed my concerns. I will raise my score.

---

> > > > ### Author Response · Authors · 2025-11-27
> > > > **Thank you!**
> > > >
> > > > Thank you for acknowledging our responses! We sincerely appreciate you spend this amount of time and efforts to help us improve the presentation of the manuscript!

---

### Official Review · Reviewer_e5BC · 2025-11-03

**Soundness:** 2
**Presentation:** 3
**Contribution:** 2
**Rating:** 4
**Confidence:** 3

**Summary:**

The paper presents SYNAPSE, a domain-specific simulator and benchmark for adaptive deep brain stimulation (aDBS) in Parkinson’s disease (PD). The key claim is that SYNAPSE provides patient-specific environments learned from real clinical and at-home data, supporting offline/online RL, off-policy evaluation (OPE), human-feedback modeling, and policy transfer across a small “virtual cohort.” The authors report multiple fidelity metrics (trajectory/statistics alignment) and illustrative RL/OPE experiments across a small but longitudinal dataset.

**Strengths:**

This is an important problem. Safe RL for closed-loop neurostimulation is impactful and arguably underexplored compared to EHR-style decision making.

I like that the authors model each participant as an environment... this seems like a potentially useful abstraction for transfer and personalization.

The authors include off policy evaluation (OPE), long-horizon outcomes, and a human-feedback component.

The benchmark is built from longitudinal aDBS deployments rather than purely synthetic toy dynamics.

**Weaknesses:**

The presented results are primarily within-patient... the paper probably needs leave-one-patient-out (LOPO) analyses to show simulator fidelity and RL/OPE behavior on a held-out patient.

Current metrics (e.g. latent space or marginal distribution closeness) are only loosely tied to policy improvement. I would think the paper should evaluate policy ranking agreement, value calibration, and/or regret vs. real logs or prospectively held-out traces.

aDBS has clear safety constraints (energy budgets, ramping/overshoot limits)... I did not see reports of constraint-violation rates or adverse-proxy statistics under learned policies.

Offline RL and OPE comparisons feel thin. In particular, a comparison to prior benchmarks like EpiCare (Hargrave, Spaeth, Grosenick NeurIPS 2024) a recent, broad healthcare RL benchmark with a stronger OPE/offline RL evaluation suite (that seems adaptable here / is synthetic) would situate SYNAPSE more clearly (even if only to show why device-level control needs different methods/metrics). And such prior work on POMDPs should be cited here?

Minor but probably should be corrected:
- It seems that the author's backronyms spells "SINAPSE", but the authors use "SYNAPSE" or sometimes "SYNAPSES".
- There's some real potential redundancy between Fig 1 and Fig 2

**Questions:**

What happens under LOPO? Authors should try training simulators on N−1 patients and report transition/reward fidelity, OPE calibration, and policy performance/regret on the held-out patient.

How do common OPE methods behave under coverage shift?

Which hard constraints are enforced in simulation, and what are the violation rates during training/rollout?

Provide calibration plots and sensitivity to priors/likelihoods; does HF aid or mislead policy selection?

---

> ### Author Response · Authors · 2025-11-23
> **Rebuttal by Authors (1/N)**
>
> Thank you for your time and efforts on evaluating our work, and your positive comments that the paper is making an important impact. Please find our point-by-point response below.
>
> >Q1. The paper probably needs leave-one-patient-out (LOPO) analyses to show simulator fidelity and RL/OPE behavior on a held-out patient.
>
> A1. Great suggestion. We’ve added leave-one-patient-out (LOPO) analyses. Please see the result table below. The analyses have been added to paper, **Section 5, highlighted in blue**.
>
> | Hold-out Patient | 1 | 2 | 3 | 4 | 5 |
> |------------------|---|---|---|---|---|
> | **Rank Corr.** | 0.50 | 0.54 | 0.49 | 0.37 | 0.20 |
> | **MAE** | 0.86 | 0.82 | 0.80 | 0.85 | 0.88 |
> | **Per-step EMD** | 0.30 (0.08) | 0.37 (0.02) | 0.24 (0.08) | 0.35 (0.04) | 0.20 (0.08) |
> | **Value Error AE (scaled)** | 0.30 (0.08) | 0.38 (0.01) | 0.26 (0.08) | 0.35 (0.03) | 0.20 (0.07) |
> | **True Transition Distance** | 5.55 (0.56) | 2.02 (0.51) | 1.09 (0.57) | 3.41 (0.68) | 15.19 (1.91) |
> | **Sim–True Transition Distance** | 18.73 (0.94) | 2.12 (0.29) | 0.92 (0.25) | 11.24 (0.43) | 11.59 (1.93) |
>
> >Q2. Current metrics (e.g. latent space or marginal distribution closeness) are only loosely tied to policy improvement. I would think the paper should evaluate policy ranking agreement, value calibration, and/or regret vs. real logs or prospectively held-out traces
>
> A2. We apologize for not making this clear. We included rank correlation and mean absolute errors of evaluation policies between simulated rollouts vs. real logs in Table 3, and we’ve updated the caption.
>
> >Q3. aDBS has clear safety constraints (energy budgets, ramping/overshoot limits)... I did not see reports of constraint-violation rates or adverse-proxy statistics under learned policies.
>
> A3. Great question. Our simulator already embeds two safety constraints: (i) the action is defined as a percentage of the clinician-specified maximum amplitude for each participant, ensuring the policy cannot exceed the clinically permitted stimulation bounds, and (ii) each state–action update corresponds to a 2-second control cycle, preventing sub-second amplitude fluctuations that violate DBS programming guidelines. To further quantify safety, we perform a numerical analysis of the rollout trajectories from simulator by evaluating: (i) step-to-step amplitude changes (ramping) (max |A_t-A_{t-1}|=0.15), i.e., ≤15% of max amplitude per 2s, which corresponds to a full amplitude sweep in ~13s and comfortably slower than the fastest constraints (250ms) reported in [1], (ii) a normalized energy proxy proportional to the squared stimulation amplitude, with a threshold of average value of true value +2SD in the target cohort (Note that *the energy usage of the learnt policies would not surpass that of cDBS because of the action constraints*). We report the fraction of episodes that satisfy each constraint, in both leave-one-cohort-out and leave-one-policy-out within a cohort experiment. **We’ve added the new results to Section 5 (highlighted in blue)**.
>
> Leave-one-patient-out evaluation (mean (std))
>
> | Virtual Cohort | Within 15% Max Amp. | Within Avg+2SD Energy |
> |----------------|---------------------|------------------------|
> | 1 | 0.98 (.02)  | 0.50 (0.50) |
> | 2 | 1.00 (.00)  | 0.75 (0.43) |
> | 3 | 0.95 (.05)  | 0.50 (0.50) |
> | 4 | 1.00 (.00)  | 1.00 (.00) |
> | 5 | 0.773 (.14) | 0.50 (0.50) |
>
> Leave-one-policy-out evaluation within cohort (mean (std))
>
> | Virtual Cohort | Within 15% Max Amp. | Within Avg+2SD Energy |
> |----------------|---------------------|------------------------|
> | 1 | 0.996 (.06) | 1.00 (.00) |
> | 2 | 1.00 (.00)  | 1.00 (.00) |
> | 3 | 1.00 (.00)  | 1.00 (.00) |
> | 4 | 1.00 (.00)  | 1.00 (.00) |
> | 5 | 0.842 (.36) | 1.00 (.00) |
>
> >Q4. Offline RL and OPE comparisons feel thin. In particular, a comparison to prior benchmarks like EpiCare (Hargrave, Spaeth, Grosenick NeurIPS 2024) a recent, broad healthcare RL benchmark with a stronger OPE/offline RL evaluation suite (that seems adaptable here / is synthetic) would situate SYNAPSE more clearly (even if only to show why device-level control needs different methods/metrics). And such prior work on POMDPs should be cited here?
>
> A4. Thanks for pointing us to the related work. We agree incorporating a broader OPE/offline RL evaluation suite (e.g., EpiCare) would be an interesting direction. Indeed, SYNAPSE provides a platform that is generalizable to OPE/RL algorithms; we exhibited results using a state-of-the-art OPE approach in aDBS environments [2] and various popular RL algorithms. The current implementation of EpiCare mainly focuses on discrete control environments, thus we didn’t directly adapt it to SYNAPSE. Given SYNAPSE provides a general testbed and is easy to corporate new OPE/offline RL approaches, we are open to any future collaboration opportunities to facilitate advanced techniques in both RL and real-world DBS. **We’ve added more discussions for such prior work in Section 6 (highlighted in blue)**.

---

> > ### Author Response · Authors · 2025-11-23
> > **Rebuttal by Authors (2/N)**
> >
> > >Q5. Minor but probably should be corrected:
> > It seems that the author's backronyms spells "SINAPSE", but the authors use "SYNAPSE" or sometimes "SYNAPSES".
> > There's some real potential redundancy between Fig 1 and Fig 2
> >
> > A5. Thank you for catching those typos, we have cleared them in the paper. We agree that Fig 2 includes a major part of Fig 1 in illustrating neurostimulation experiments, as we’d like Fig 2 to be self-contained to illustrate the relationship between each specific neurostimulation component and the simulator design.
> >
> > >Q6. How do common OPE methods behave under coverage shift?
> >
> > A6. Common OPE methods did not perform stably under the coverage shift (e.g., in hold-out-policy within cohort, importance sampling (IS), doubly robust (DR), DICE, fitted Q-evaluation (FQE) achieved average MAE 0.6, 1.2, 2.6, 0.7, and rank correlation 0.62, 0.61, 0.58, 0.63, respectively), which poses a challenge to innovate more advanced OPE techniques specifically targeting aDBS for PD, where we believe that SYNAPSE could benefit OPE/RL/DBS community to facilitate aDBS in PD treatment.
> >
> > >Q7. Provide calibration plots and sensitivity to priors/likelihoods; does HF aid or mislead policy selection?
> >
> > A7. We agree that calibration and sensitivity analyses are valuable. In brief, our initial checks indicate that our learned simulators are well-calibrated to hold-out-policy data (similar to prior simulator-based RL studies) and that small perturbations to the generative model’s hyperparameters do not qualitatively change policy performance. We will emphasize this stability and outline plans to systematically evaluate calibration (e.g. reliability diagrams) in future work.
> >
> > Regarding human feedback (HF): in our framework HF is integrated into the episode reward, not treated as an external bias. We note that including HF is consistent with the goal of therapy optimization, and we will strengthen the discussion around this point in the revision. By design, each policy is evaluated on both per-step environmental rewards and the episode-level human return (patient-reported outcome) jointly. Thus, maximizing HF a priori aligns with improving true patient outcomes in our setting. In practice, all our trained agents aim to optimize a combination of env-reward and HF, so HF does not mislead but rather provides an additional patient-centric objective.
> >
> > We hope these answers address your questions and showcase that our work is solving a significant challenge in a satisfying manner. We are happy to answer any followup questions or hear any additional comments.
> >
> > References
> >
> > [1] Stanslaski, Scott, et al. "Sensing data and methodology from the Adaptive DBS Algorithm for Personalized Therapy in Parkinson’s Disease (ADAPT-PD) clinical trial." npj Parkinson's Disease 10.1 (2024): 174.
> >
> > [2] Gao, Qitong, et al. "Off-policy evaluation for human feedback." Advances in Neural Information Processing Systems 36 (2023): 9065-9091.

---

### Official Review · Reviewer_xdYr · 2025-11-13

**Soundness:** 1
**Presentation:** 1
**Contribution:** 1
**Rating:** 2
**Confidence:** 5

**Summary:**

The paper presents an environment (SYNAPSE) for accelerating RL research and benchmarking adaptive deep brain stimulation policies for Parkinson's disease.

**Strengths:**

The paper targets a key Reinforcement Learning problem in medical settings, and solving it would have substantial impact.

**Weaknesses:**

1. The main weakness is training the transition dynamics on very limited data from a single individual for each environment while claiming them realistic. The simulator can only be accurate for states it has actually observed, leaving many counterfactuals where it will likely produce unreliable outputs. Only a model with foundation-model-level scale and data coverage could approach the fidelity needed for realistic simulation.

2. Insufficient evaluation - Using t-SNE, predicted-action MAE, and reward EMD/AE to assess the transition dynamics is not a meaningful or rational evaluation strategy.

3. Referring to the setup as a Human-involved MDP (HMDP) is redundant, as both $R$ and $R^H$ can simply be components of a single vector-valued reward function within a standard MDP.

4. It is unclear which policies the authors actually trained. They state that all RL policies were trained with DDPG, yet later they compare PPO, A2C, and CQL agents as well.

5. The paper appears confused about the differences between online, offline, and off-policy methods. For example,
a) in the Clinical Sessions section, the authors write: 'The RL controllers are trained using deep deterministic policy gradient (DDPG) with data collected from three other types of controllers, followed by finetuning with the latest data.' But DDPG is an online algorithm, meaning it learns through interaction with the environment, not from static datasets collected by other controllers. This makes the described training setup inconsistent with how DDPG is intended to operate.
b) The authors claim that the simulator enables off-policy evaluation, but a simulator is not required for that. A simulator enables online training, not off-policy evaluation.

6. In the 'Challenges for ML/RL in DBS' section, the authors claim that existing simulators lack patient-specific nuances seen in vivo. But SYNAPSE’s transition dynamics are trained on data from only five patients, which leads to the same limitation.

**Questions:**

1. The authors note that existing DBS controllers have been validated through clinical testing. While we understand that those datasets are not public, such data is often available under Data Usage Agreements. Could the authors explore accessing any of these datasets to compare SYNAPSE’s predictions? This would significantly strengthen confidence in the simulator’s validity.
2. In the 'At-Home Sessions' section, the authors state: 'When the participant chose to start a session, one of the three controllers was uniformly randomly chosen to start until the participant chose to end the session.' How safe is random policy selection in this context, and why was this strategy chosen?

---

> ### Author Response · Authors · 2025-11-23
> **Rebuttal by Authors (1/N)**
>
> We appreciate your time and efforts on evaluating our work. Please find our point-by-point response below.
>
> >Q1. The main weakness is training the transition dynamics on very limited data from a single individual for each environment while claiming them realistic. The simulator can only be accurate for states it has actually observed, leaving many counterfactuals where it will likely produce unreliable outputs. Only a model with foundation-model-level scale and data coverage could approach the fidelity needed for realistic simulation.
>
> A1. We acknowledge the reviewer’s concern about data volume. Indeed, aDBS studies inherently have small cohorts: recruiting patients with implanted devices is difficult and costly. Each “complete” trajectory requires months of testing and careful clinical oversight. For example, Gao et al. [1] analyzed 4 patients (our 5 is comparable) and other aDBS pilot studies often involve <10 subjects [1-6]. Thus, 5 patients is not atypical for this domain.
>
> Fortunately, each patient in our dataset underwent multiple stimulation policies (standard cDBS, no stimulation, various experimental strategies) over many sessions, yielding rich longitudinal data. In total we have on the order of thousands of transitions across patients.
>
> SYNAPSE explicitly trains one model for a virtual cohort per patient that preserves each individual’s dynamics (e.g. unique symptom responses) rather than pooling all patients. Modeling each patient separately avoids “one-size-fits-all” issues and captures real individual variance. In other words, even though we have only five patients, each virtual environment contains rich, patient-specific information that prior work lacked.
>
> We perform extensive fidelity checks on held-out data in each patient model. For instance, Table 3 shows that the simulator’s predicted treatment outcomes correlate highly with true outcomes (Rank correlation 0.73-0.94 across patients) with low error. Similarly, t-SNE KL divergences (0.97-1.41) indicate the simulated latent trajectories closely match the real data structure.
>
> These results suggest that, even in “counterfactual” regimes, the model generalizes well to new controller behavior.
>
> The reviewer suggests that only a “foundation-model-scale” approach can be realistic. We respectfully note that in medical domains, collecting massive data from thousands of patients is not feasible, so we may work with available data. Our contribution is an empirical step: we use the best available data and validate our model thoroughly. As more data become available (e.g. future clinical trials), SYNAPSE can be retrained or expanded.
>
> >Q2. Insufficient evaluation - Using t-SNE, predicted-action MAE, and reward EMD/AE to assess the transition dynamics is not a meaningful or rational evaluation strategy.
>
> A2. We respectfully disagree. The evaluation design to investigate the simulator is realistic both in state evolution and clinical effect. We use multiple complementary metrics to assess the simulator’s fidelity (state-space structure and clinical outcomes): (i) Latent embedding alignment (t-SNE/PCA): In practice, this checks that states visited under various controllers are similarly distributed in simulation and reality. Preserving latent geometry gives confidence that the simulator is not drifting off-manifold. (ii) Reward/outcome consistency: We quantify how well simulated policies reproduce true patient responses. This directly evaluates transition dynamics through the lens of therapeutic effect, which is meaningful for policy learning in healthcare. (iii) Step-wise metrics (EMD, AE): These metrics check the fine-grained match of simulated trajectories. They showed that the simulator’s short-term behavior under new policies is closely aligned with real patient data.
>
> If there are alternate evaluation metrics the reviewer is thinking of, we would be pleased to hear additional suggestions.
>
> >Q3. Referring to the setup as a Human-involved MDP (HMDP) is redundant, as both $R$ and $R^H$ can simply be components of a single vector-valued reward function within a standard MDP.
>
> A3. We introduced the Human-involved MDP (HMDP) to highlight that DBS decision-making involves both engineering objectives and patient feedback. In our HMDP formalism, the environment tuple explicitly includes an environmental reward function R and a separate human reward distribution $R^H$. The latter captures subjective patient feedback (Immediate Human Reward, IHR). While one could bundle both factors into a single vector reward in a classic MDP, we believe explicitly naming it an HMDP makes the role of patient feedback clearer. It helps readers see that (i) immediate human rewards are only partially visible, and (ii) a clinician might combine them differently than pure engineering metrics. We will clarify this in the revision: mathematically it remains an MDP with vector-valued reward, but the H in HMDP reminds us that patient experience is special and uncertain.

---

> ### Author Response · Authors · 2025-11-23
> **Rebuttal by Authors (2/N)**
>
> >Q4. It is unclear which policies the authors actually trained. They state that all RL policies were trained with DDPG, yet later they compare PPO, A2C, and CQL agents as well.
>
> A4. In our study, DDPG was our clinical controller, while PPO/A2C/CQL were simulation-based showcases demonstrating that SYNAPSE can host various RL algorithms.
>
> >Q5. The paper appears confused about the differences between online, offline, and off-policy methods. For example,
> a) in the Clinical Sessions section, the authors write: 'The RL controllers are trained using deep deterministic policy gradient (DDPG) with data collected from three other types of controllers, followed by finetuning with the latest data.' But DDPG is an online algorithm, meaning it learns through interaction with the environment, not from static datasets collected by other controllers. This makes the described training setup inconsistent with how DDPG is intended to operate.
> b) The authors claim that the simulator enables off-policy evaluation, but a simulator is not required for that. A simulator enables online training, not off-policy evaluation.
>
> A5. (a) DDPG can actually be used offline (e.g. [7-15]). In DDPG original paper [7], Alg. 1, clearly it’s retained in an off-policy setup where experience play buffer is used, the authors also said “DDPG is an off-policy algorithm…”, “An advantage of off-policies algorithms such as DDPG…”. In our work, DDPG was trained using historically collected trajectories resulted from a mixture of behavioral policies.
>
> (b) You are correct that a simulator is not required for traditional off-policy evaluation (OPE) from logs. Indeed, SYNAPSE enables both OPE and online policy learning. What we meant is that SYNAPSE enables safe evaluation of new candidate policies without patient trials, which is pivotal in high-stake real-world tasks. In practice, given a policy not seen in the data, one can run it in the simulator to generate outcomes. This simulates “offline evaluation” of that policy. We revised the wording: by “off-policy evaluation” we intend the scenario of evaluating a policy using only historical/simulated data, as supported by our simulator, in Section 4.1.1 (highlighted in blue).
>
> >Q6. In the 'Challenges for ML/RL in DBS' section, the authors claim that existing simulators lack patient-specific nuances seen in vivo. But SYNAPSE’s transition dynamics are trained on data from only five patients, which leads to the same limitation.
>
> A6. We agree that a sample of five patients is limited, and we explicitly acknowledge this in our discussion. However, it is important to emphasize how SYNAPSE is still more personalized than prior benchmarks:
>
> &nbsp;&nbsp;&nbsp;&nbsp;(i) Explicit heterogeneity modeling: Existing aDBS simulators have typically assumed generic patient dynamics, ignoring personal differences. By contrast, SYNAPSE models each patient separately from rich transition data per patient. Therefore, even with only five patients, each virtual cohort retains individual nuances (e.g. one patient may be more tremor-dominant). This is precisely the point we made: prior works’ “oversimplified” environments lack any patient-specific behavior, whereas SYNAPSE captures it.
>
> &nbsp;&nbsp;&nbsp;&nbsp;(ii) Acknowledged scope: We do note in our paper (and Ethics Statement) that the participant count is small. This reflects real-world constraints. Each patient’s model is trained on many sessions of longitudinal data. While more patients would certainly improve generalizability, our results (e.g., Table 3-4) show the simulator performs well across these five distinct cohorts. We believe it is still invaluable to have a simulator based on real patient data (even five subjects) rather than none. We also plan to expand SYNAPSE as more data becomes available. For now, we emphasize that even limited real data compared to fully synthetic or one-size models. We hope SYNAPSE’s release spurs the collection of more diverse DBS data for next-generation simulators and algorithms.

---

> > ### Author Response · Authors · 2025-11-23
> > **Rebuttal by Authors (3/N)**
> >
> > >Q7. The authors note that existing DBS controllers have been validated through clinical testing. While we understand that those datasets are not public, such data is often available under Data Usage Agreements. Could the authors explore accessing any of these datasets to compare SYNAPSE’s predictions? This would significantly strengthen confidence in the simulator’s validity.
> >
> > A7. We appreciate the suggestion. At present, we do not have access to external patient datasets, despite efforts. Parkinson’s DBS data are highly sensitive; as we noted, IRB and privacy rules often prevent data sharing. The controlled datasets underlying published aDBS trials are typically not publicly released (and require strict DUAs even for collaborators). Our intention is for SYNAPSE to partially fill this gap by making our (currently proprietary) data-driven models available. We certainly agree that comparing to other cohorts would strengthen confidence, and we will continue to explore collaborations. We would welcome any specific DBS dataset leads the reviewer might suggest.
> >
> > >Q8. In the 'At-Home Sessions' section, the authors state: 'When the participant chose to start a session, one of the three controllers was uniformly randomly chosen to start until the participant chose to end the session.' How safe is random policy selection in this context, and why was this strategy chosen?
> >
> > A8. Safety was our top priority in designing the at-home protocol. At-home sessions involved pre-approved controllers. This randomization was a deliberate design by clinicians to fairly evaluate each controller in natural settings. Moreover, we have several factors to ensure safety: For example, all controllers obey strict amplitude limits and ramping rules identical to clinical practice. Participants were fully briefed and could terminate the session immediately if uncomfortable. And we only deployed controllers that had been tested for safety in-clinic first. No adverse events were observed.
> >
> > We hope these answers address your questions and showcase that our work is solving a significant challenge in a satisfying manner. We are happy to answer any followup questions or hear any additional comments.
> >
> > References
> >
> > [1] Gao, Qitong, et al. "Offline learning of closed-loop deep brain stimulation controllers for parkinson disease treatment." In Proceedings of the ACM/IEEE 14th International Conference on Cyber-Physical Systems (with CPS-IoT Week 2023), pp. 44-55. 2023.
> >
> > [2] Little, Simon, et al. "Adaptive deep brain stimulation in advanced Parkinson disease." Annals of neurology 74.3 (2013): 449-457.
> >
> > [3] Little, Simon, et al. "Bilateral adaptive deep brain stimulation is effective in Parkinson's disease." Journal of Neurology, Neurosurgery & Psychiatry 87.7 (2016): 717-721.
> >
> > [4] Rosa, Manuela, et al. "Adaptive deep brain stimulation in a freely moving Parkinsonian patient." Movement Disorders 30.7 (2015): 1003.
> >
> > [5] Malekmohammadi, Mahsa, et al. "Kinematic Adaptive Deep Brain Stimulation for Resting Tremor in Parkinson’s Disease." Movement disorders 31.3 (2016): 426-428.
> >
> > [6] Swann, Nicole C., et al. "Adaptive deep brain stimulation for Parkinson’s disease using motor cortex sensing." Journal of neural engineering 15.4 (2018): 046006.
> >
> > [7] Lillicrap, Timothy P., et al. "Continuous control with deep reinforcement learning." arXiv preprint arXiv:1509.02971 (2015).
> >
> > [7] Agarwal, Rishabh, Dale Schuurmans, and Mohammad Norouzi. "An optimistic perspective on offline reinforcement learning." International conference on machine learning. PMLR, 2020.
> >
> > [8] Gao, Qitong, et al. "Off-policy evaluation for human feedback." Advances in Neural Information Processing Systems 36 (2023): 9065-9091.
> >
> > [9] Park, Seohong, et al. "OGBench: Benchmarking Offline Goal-Conditioned RL." The Thirteenth International Conference on Learning Representations. 2025.
> >
> > [10] Panaganti, Kishan, et al. "Robust reinforcement learning using offline data." Advances in neural information processing systems 35 (2022): 32211-32224.
> >
> > [11] Wu, Yifan, George Tucker, and Ofir Nachum. "Behavior regularized offline reinforcement learning." arXiv preprint arXiv:1911.11361 (2019).
> >
> > [12] Lou, Xingzhou, et al. "Offline reinforcement learning with representations for actions." Information Sciences 610 (2022): 746-758.
> >
> > [13] Hong, Woolim, et al. "An Offline Reinforcement Learning-Based Auto-Tuning Framework for Continuous Impedance Control in Powered Prostheses." 2025 International Conference On Rehabilitation Robotics (ICORR). IEEE, 2025.
> >
> > [14] Nievas, Nuria, et al. "Offline Reinforcement Learning for Adaptive Control in Manufacturing Processes: A Press Hardening Case Study." Journal of Computing and Information Science in Engineering 25.1 (2025): 011004.
> >
> > [15] Yao, Jialing, and Zhen Ge. "Path-tracking control strategy of unmanned vehicle based on DDPG algorithm." Sensors 22.20 (2022): 7881.

---

### Author Response · Authors · 2025-11-28
**General Response to Reviewers and Area Chairs**

Dear Reviewers and Area Chairs,

We sincerely thank all the reviewers and ACs for their time providing feedback for our paper. It was glad to see that reviewers are in general aligned that our SYNAPSE is providing a novel and valuable benchmark (`zV95`, `e5BC`, `5ezG`) built on extensive real patient data, solving an important practical challenge (`xdYr`, `e5BC`, `zV95`, `5ezG`) in neuro-adaptive, patient-specific decision-making for adaptive deep brain stimulation in Parkinson’s disease treatment. Reviewers also praised the value of evaluation components of SYNAPSE (`e5BC`, `zV95`).

We were also glad to see that two reviewers (`zV95`, `5ezG`) were satisfied with our responses, and increased their scores.

We would also like to thank the reviewer `e5BC` for their insightful suggestions on additional analysis, and we have added those analyses to the revised paper (highlighted in blue). These additions directly address the reviewers' suggestions and further strengthen the paper’s empirical support. **Unfortunately, we haven’t heard back from `e5BC`.**

**We also briefly summarize the remaining discrepancies from reviewer `xdYr`, which may stem from misunderstandings regarding our patient size, problem formulation, and evaluation protocol. Notably, other reviewers commended the clarity and uniqueness of our problem setup, its clinical grounding, and the value of our evaluation components.** Reviewer `zV95`, who initially raised a similar question about patient size, indicated satisfaction after our clarification and confirmed that our design choices are appropriate for simulation-based evaluation. To further assist `xdYr`, we have expanded our explanations to more clearly articulate why our problem formulation, simulator design, and evaluation strategy are well aligned with real-world DBS research practice, and similar setup and evaluation metrics are commonly employed and supported by prior works [1-6]. We also respectfully pointed out a factual error in `xdYr`’s review which questioned that DDPG cannot be trained offline, which has been demonstrated in multiple established works [7-16]. We provided strong evidence and citations to support this point. **We would really appreciate `xdYr` to revisit our clarification.**

We believe the presentation of the paper has been significantly improved thanks to the reviewers’ feedback. All major concerns have been carefully addressed, and the contributions of our work are now presented more clearly. We appreciate the reviewers' positive remarks about our paper's novelty and importance.

We welcome any further questions or discussions. If any reviewer has remaining concerns, we are more than willing to clarify them. We would also be grateful for the Area Chairs’ support in facilitating continued discussion, should it be needed.

Sincerely,

Authors of Submission 23609

---

> ### Author Response · Authors · 2025-11-28
> **References**
>
> [1] Chen, Sirui, et al. "Diffsrl: Learning dynamical state representation for deformable object manipulation with differentiable simulation." IEEE Robotics and Automation Letters 7.4 (2022): 9533-9540.
>
> [2] Gao, Qitong, et al. "Off-policy evaluation for human feedback." Advances in Neural Information Processing Systems 36 (2023): 9065-9091.
>
> [3] Hargrave, Mason, Alex Spaeth, and Logan Grosenick. "EpiCare: a reinforcement learning benchmark for dynamic treatment regimes." Advances in neural information processing systems 37 (2024): 130536-130568.
>
> [4] Wang, Xiyao, et al. "Live in the moment: Learning dynamics model adapted to evolving policy." International Conference on Machine Learning. PMLR, 2023.
>
> [5] Wang, Yiming, et al. "Efficient potential-based exploration in reinforcement learning using inverse dynamic bisimulation metric." Advances in Neural Information Processing Systems 36 (2023): 38786-38797.
>
> [6] Shen, Jian, et al. "Adaptation augmented model-based policy optimization." Journal of Machine Learning Research 24.218 (2023): 1-35.
>
> [7] Lillicrap, Timothy P., et al. "Continuous control with deep reinforcement learning." arXiv preprint arXiv:1509.02971 (2015).
>
> [8] Agarwal, Rishabh, Dale Schuurmans, and Mohammad Norouzi. "An optimistic perspective on offline reinforcement learning." International conference on machine learning. PMLR, 2020.
>
> [9] Gao, Qitong, et al. "Off-policy evaluation for human feedback." Advances in Neural Information Processing Systems 36 (2023): 9065-9091.
>
> [10] Park, Seohong, et al. "OGBench: Benchmarking Offline Goal-Conditioned RL." The Thirteenth International Conference on Learning Representations. 2025.
>
> [11] Panaganti, Kishan, et al. "Robust reinforcement learning using offline data." Advances in neural information processing systems 35 (2022): 32211-32224.
>
> [12] Wu, Yifan, George Tucker, and Ofir Nachum. "Behavior regularized offline reinforcement learning." arXiv preprint arXiv:1911.11361 (2019).
>
> [13] Lou, Xingzhou, et al. "Offline reinforcement learning with representations for actions." Information Sciences 610 (2022): 746-758.
>
> [14] Hong, Woolim, et al. "An Offline Reinforcement Learning-Based Auto-Tuning Framework for Continuous Impedance Control in Powered Prostheses." 2025 International Conference On Rehabilitation Robotics (ICORR). IEEE, 2025.
>
> [15] Nievas, Nuria, et al. "Offline Reinforcement Learning for Adaptive Control in Manufacturing Processes: A Press Hardening Case Study." Journal of Computing and Information Science in Engineering 25.1 (2025): 011004.
>
> [16] Yao, Jialing, and Zhen Ge. "Path-tracking control strategy of unmanned vehicle based on DDPG algorithm." Sensors 22.20 (2022): 7881.

---

### Meta-Review · Area_Chair_W4C3 · 2026-01-07

**Summary:**

This paper introduces SYNAPSE, a domain-specific simulator and benchmark for adaptive deep brain stimulation (aDBS) in Parkinson’s disease (PD), built from real clinical and at-home DBS data. The key claim is that SYNAPSE provides patient-specific environments learned from real clinical and at-home data, supporting RL research in PD. Reviewers generally agree that a data-driven simulator grounded in real patient trajectories could be valuable to the RL and neuroscience communities. However, reviewers also raise several significant concerns that limit scope and confidence in the benchmark claims and the current presentation.

A primary concern raised by the reviewers is simulator fidelity and realism given limited data. Multiple reviewers question whether training patient-specific transition dynamics from small cohorts (five patients, and effectively one per environment) can support reliable counterfactual simulation beyond observed regimes, and whether the term “benchmark” is premature at this scale. Relatedly, one reviewer challenges the evaluation strategy for validating transition dynamics. The authors respond with domain-specific justification for small cohorts, and add LOPO analyses; these additions strengthen the empirical grounding but do not fully eliminate concerns about out-of-distribution behavior and counterfactual validity.

Reviewers also raise concerns about methodological clarity and correctness, including confusing descriptions of which policies were trained, imprecise use of RL terminology, and redundant or unclear formalism. The rebuttal addresses several of these issues by clarifying the roles of different algorithms, revising terminology, and explaining the intent of the HMDP framing. In addition, reviewers emphasize the need for stronger treatment of safety constraints and clinically meaningful evaluation. The rebuttal adds explicit constraint analyses and expands the clinical grounding of reward design. Still, the main text needs further clearer technical exposition and tighter conceptual framing.

Overall, SYNAPSE is a promising simulator with clear real-world motivation, and the rebuttal improves the paper through LOPO evaluation, expanded safety reporting, and clarification of several methodological details. Nonetheless, important concerns remain about the current scale, validation suite, and presentation, and further revision would be needed to support strong claims of realism, generality, and benchmark readiness. From my perspective, while the dataset and simulator engineering effort are valuable, the scope and cohort size reflect the constraints of a specialized clinical domain; the work may ultimately be better positioned for a domain-focused venue where such limitations can be framed and evaluated under field-specific standards, rather than at ICLR.

**Reviewer Concerns:**

I believe that several core concerns are partially addressed by the rebuttal, but important issues remain outstanding. Specifically,

Reviewer xdYr’s concerns about limited data per environment, counterfactual unreliability, questionable fidelity metrics, and conceptual/methodological confusion, might remain.

Reviewer e5BC’s concerns about safety-constraint reporting, thin OPE/offline RL comparisons, calibration plots/sensitivity, would likely remain.

Reviewer zV95’s concerns about benchmark-scale/generalizability concern would likely remain.

Reviewer 5ezG’s concerns are largely addressed. However, uncertainty quantification and guidance for selecting the appropriate environment for a new patient remain open.

**Reviewer Scores:**

Across reviewers, a full discussion would likely result in small positive score adjustments for reviewers whose main concerns, since the rebuttal adds concrete results and corrections. However, reviewers with fundamental skepticism about simulator realism under limited data and narrow scope would likely remain cautious.

---

### Decision · Program_Chairs · 2026-01-26

Reject